# A molecular staging model for accurately dating the endometrial biopsy

W. T. Teh ®[1,2,3,13], J. Chung[1,4,13], S. J. Holdsworth-Carson[1,2,5], J. F. Donoghue[1,2], M. Healey[1,2], H. C. Rees[2,6], S. Bittinger[2,6], V. Obers[7], C. Sloggett[4,8], R. Kendarsari[9,10], J. N. Fung[11], S. Mortlock ®[9], G. W. Montgomery ®[9], J. E. Girling ®[1,12] & P. A. W. Rogers ®[1,2] ✉

Natural variability in menstrual cycle length, coupled with rapid changes in endometrial gene expression, makes it difficult to accurately define and compare different stages of the endometrial cycle. Here we develop and validate a method for precisely determining endometrial cycle stage based on global gene expression. Our 'molecular staging model' reveals significant and remarkably synchronised daily changes in expression for over 3400 endometrial genes throughout the cycle, with the most dramatic changes occurring during the secretory phase. Our study significantly extends existing data on the endometrial transcriptome, and for the first time enables identification of differentially expressed endometrial genes with increasing age and different ethnicities. It also allows reinterpretation of all endometrial RNA-seq and array data that has been published to date. Our molecular staging model will significantly advance understanding of endometrial-related disorders that affect nearly all women at some stage of their lives, such as heavy menstrual bleeding, endometriosis, adenomyosis, and recurrent implantation failure.

The endometrium plays an essential role in embryo implantation, placentation and success or otherwise of pregnancy in all mammals. A fundamental understanding of human endometrial biology underpins our knowledge of everyday physiological and pathological processes that include uterine receptivity, pregnancy, menstruation, heavy menstrual bleeding, recurrent implantation failure, endometriosis, adenomyosis, endometrial cancer and pelvic pain. Nearly all women during their lifetime will see their gynaecologist for one or more endometrial-related health problems[1]. Despite this impact on quality of life for most women, the endometrium remains understudied relative to the healthcare burden of endometrial-related disorders, with

endometriosis[2], menstrual problems[1] and contraceptive-related bleeding issues[3] as prime examples.

There are two methodological challenges that have a profound impact on endometrial research: the large normal variation in menstrual cycle length, and the huge variability in gene expression across the cycle. Compared to other tissues in the body, the endometrium undergoes dramatic cyclical changes in gene expression[4,5]. Daily and sometime hourly changes in expression are driven by increased circulating estrogen from the developing ovarian follicles during the proliferative phase of the cycle, then by progesterone from the corpus luteum following ovulation, and in non-conception cycles, during

[1]University of Melbourne Department of Obstetrics and Gynaecology, Melbourne, Victoria, Australia. [2]Royal Women's Hospital, Melbourne, Victoria, Australia. [3]Melbourne IVF, Melbourne, Victoria, Australia. [4]Melbourne Bioinformatics, University of Melbourne, Melbourne, Victoria, Australia. [5]Julia Argyrou Endometriosis Centre, Epworth HealthCare, Richmond, Victoria, Australia. [6]Royal Children's Hospital, Melbourne, Victoria, Australia. [7]Melbourne Pathology, Collingwood, Victoria, Australia. [8]Microbiological Diagnostic Unit Public Health Laboratory, Department of Microbiology and Immunology, University of Melbourne at the Peter Doherty Institute, Melbourne, Victoria, Australia. [9]Institute for Molecular Bioscience, University of Queensland, St Lucia, Queensland, Australia. [10]Illumina Inc. 11 Biopolis Way, Singapore 138667, Singapore. [11]School of Biomedical Sciences, University of Queensland, St Lucia, Queensland, Australia. [12]Department of Anatomy, School of Biomedical Sciences, University of Otago, Dunedin, Aotearoa, New Zealand. [13]These authors contributed equally: W. T. Teh, J. Chung. ✉e-mail: parogers@unimelb.edu.au

menstruation after the demise of the corpus luteum and the loss of circulating progesterone. This rapidly changing gene expression within a highly variable length menstrual cycle has made accurate comparisons between matched samples difficult at best, and often impossible. As a consequence, studies linking endometrial gene expression to various endometrial-related pathologies such as fibroid-related heavy menstrual bleeding, reduced endometrial receptivity for implantation, and endometriosis, seldom replicate[6–9].

A critical variable in assessing differential endometrial gene expression between samples is accurate menstrual cycle staging. There is large variability between women in overall cycle length, as well as days of menstrual bleeding, and follicular and luteal phase lengths[10]. In a study of over 30,000 women, only 12.4% had a 28-day cycle[11]. Most had menstrual cycle lengths between 23 and 35 days, with a normal distribution centred on day 28, and over half had cycles that varied by 5 days or more from cycle to cycle. There was a 10-day spread of observed ovulation days for a 28-day cycle, with the most common day of ovulation being day 15. Another large study of 612,613 ovulatory cycles reported a mean length of 29.3 days from 124,648 subjects[12]. The mean follicular phase length was 16.9 days (95% CI: 10–30) and mean luteal phase length was 12.4 days (95% CI: 7–17). Part of the variability in cycle length between women was due to age, with a consistent shortening of the average cycle length by about 3 days from 30 down to 27 days between ages 25 and 45[12,13].

Methods currently in use for estimating endometrial cycle stage have limitations. Endocrine methods measuring the luteinising hormone (LH) surge or peripheral blood estrogen and progesterone are indirect and do not allow for variability over time in endometrial response. Ultrasound scans to detect follicle size and/or ovulation do not provide an obligatory correlation with endometrial development. Recording the commencement of last menstrual period (LMP) gives an accurate fix on a major endometrial event, but as a single fixed point in the cycle is of limited use for accurately comparing different stages of cycles of variable length. Histopathology of the endometrium is the most direct measure of endometrial stage and normalcy[14], although this is a subjective method with inherent inaccuracy even among experts[15]. Although significant advances have been made using endometrial gene expression to determine cycle stage, particularly in the mid-luteal phase around the time of embryo implantation[4,16,17], these methods do not cover the whole cycle.

A more precise method for normalising endometrial gene expression across the menstrual cycle will provide a major contribution to understanding endometrial function and provide foundational information to investigate the pathophysiology of common gynaecological conditions such as heavy menstrual bleeding, recurrent implantation failure and endometriosis.

Therefore, the first aim of this study was to develop and validate a new method for accurately determining menstrual cycle stage based on changing endometrial gene expression. The second aim was to demonstrate the functional utility of the new method by using normalised endometrial gene expression data to identify genes that change expression most rapidly across the menstrual cycle, as well as investigate the effects of increasing age and ancestry on differential gene expression in the endometrium. We have previously demonstrated strong genetic effects on endometrial gene expression with some evidence for genetic regulation of gene expression in a menstrual cycle stage-specific manner[18,19]. However, to date no-one has identified differentially expressed endometrial genes between women of different ancestries, despite well-established differences in genetic makeup.

## Results

### Subject Details

The median age of all subjects (study 1, 2 and Illumina HT-12 validation study) at time of endometrial biopsy was 33 years (range 18–49). Of the total of 358 subjects, 214 had confirmed endometriosis, 131 did not have endometriosis and in 13 endometriosis status was unknown. Similarly, 167 had had a prior clinical pregnancy, 183 had never been pregnant, and pregnancy status information was unavailable for the remaining 8.

The average age at time of endometrial biopsy of the 236 subjects in Study 1 from which the final molecular model was developed was 31.1 years (range 18–49). All these women provided endometrial biopsies at the time of diagnostic laparoscopy for suspected endometriosis, with the primary symptom for investigation being pelvic pain. 168 (71%) had endometriosis and 68 (29%) did not. Of the 236 subjects, 60 (25%) had had a successful prior pregnancy (Supplementary Table 1). Fertility intention information was available for 136 of the women (Supplementary Table 2). Only 28 women reported problems conceiving (defined as trying for more than 12 months to conceive), of whom 10 went on to have successful pregnancies, and only 6 out of 96 subjects reported pregnancy loss due to miscarriage, although this number could have been higher due to missing data. All subjects reported regular menstrual cycles and normal endometrium as assessed by at least one experienced pathologist.

### Analysis 1: Development of the 'molecular staging model' to assign cycle stage for secretory stage samples only

Splines were fitted to RNA-seq expression data for each of 20,067 genes from 96 endometrial samples where 2 or 3 independent pathology reports agreed to within 2 post-ovulatory days (Fig. 1, panel 1). For each endometrial sample, an estimated post-ovulatory day (POD) was obtained using the day which minimised mean squared error (MSE) between the observed expression and the expected expression across all genes. Examples of MSE plots are shown in the Fig. 1, panel 2. There was a strong correlation between the POD cycle time calculated from the lowest MSE value and the average of the pathology estimates (r = 0.9297) (Fig. 1, panel 3). To illustrate that larger, less precise, units of time can be used to estimate cycle time using the same method, an additional model was built using the pathology-assigned 3 secretory cycle stages (i.e., early-, mid-, and late-secretory). The cycle time estimated from the 3 stages model showed a strong correlation to the cycle time estimated from the 14-day POD model (r = 0.9807) (Fig. 1, panel 4).

### Analysis 2: Molecular staging model using 7 pathology stages for the whole cycle with RNA-seq and array expression data

In Analysis 2 we modelled RNA-seq expression data from all 236 samples collected in Study 1. These samples had been classified by routine pathology into 1 of 7 cycle stages. Because the majority of the proliferative phase samples were not assigned as early, mid or late by the pathologist, we re-assigned all samples labelled as proliferative into early, mid, and late by fitting a penalised cubic regression spline (k = 3) using gene expression data from samples classified by the pathologists as menstrual, proliferative, and early secretory (Fig. 2a). Then a proliferative time point was estimated from the minimised MSE between the observed expression and the expected expression across all genes (Fig. 2b). The proliferative samples were then split into equal sized groups of early, mid, and late using this time point (Fig. 2c). A penalised cyclic cubic regression spline (k = 8) was fit for all 20,067 genes using the 7 stages of the menstrual cycle, which included the re-assigned early, mid and late proliferative samples (Fig. 2d). Each endometrial sample was then assigned a 'day' or 'model time' using the time which minimised the MSE between the observed expression data for all genes and their corresponding gene models (Fig. 2e). 'Model time' is a relative timepoint in the cycle and does not correspond to a real day. Under the assumption that all 236 women were approximately uniformly distributed across the menstrual cycle, the data were transformed so that the distance in time between each sample was identical

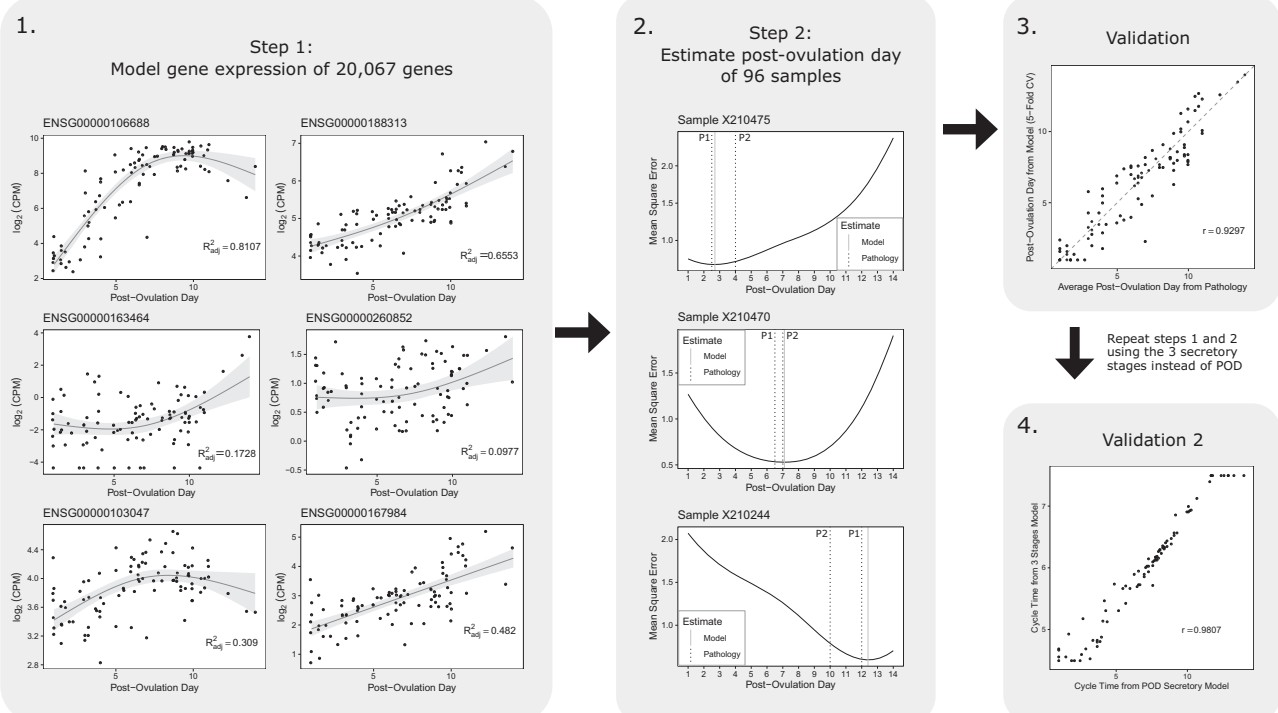

**Fig. 1 | Analysis 1: Development of the molecular staging model to assign cycle stage for secretory stage samples only.** Panel 1. Examples of regression spline fitting to expression data for individual genes from 96 endometrial samples taken between post-ovulatory days (POD) 1–14. Splines were fitted to a total of 20,067 genes. The error bands surrounding the regression lines represent the 95% confidence interval. Panel 2. Plots showing post-ovulatory time that gives lowest Mean Squared Error (MSE) using spline data for all 20,067 probes for 3 different endometrial samples (solid line). Dotted lines show POD estimates from 2 independent evaluations by experienced pathologists. Panel 3. Correlation between average POD from 2 or 3 independent pathology evaluations and the POD time at which the lowest MSE occurred. Panel 4. Correlation between the estimated POD from the POD secretory model and the estimated cycle time from the 3 stages secretory model. Source data are provided as a Source Data file.

(Fig. 2f). This ranked all the samples in order from the start to the end of the cycle, removing the need for cycle stages or an idealised 28-day cycle. At this point the x-axis was changed to show the percentage of the way through the menstrual cycle that each sample was. The new time points were also compared to the pathology-derived cycle stages to get an approximation how the model time corresponds to stages in the menstrual cycle (Fig. 2g). Gene curves were then refitted using the newly derived cycle times for each sample with a penalized cyclic cubic regression spline (k = 30) (Fig. 2h). For visualisation purposes, normalisation of gene expression for cycle stage was then derived by subtracting the expected expression from the observed expression (i.e., calculating the residuals) and re-adding the mean (Fig. 2i).

### Validation of the molecular staging model
Various validation studies were undertaken using the molecular staging model. As an initial check, data from Analysis 1 using POD to develop the secretory model was plotted against secretory stage data from the final molecular staging model generated in Analysis 2 (Fig. 3a). A second comparison confirmed that using only 3 cycle stages (early, mid and late secretory from only 1 pathologist) gave similar results to having more frequent daily POD information from 2 or 3 independent pathologists (Fig. 1, panel 4). To assess the repeatability of the molecular staging model method, Analyses 1 and 2 were repeated using Illumina HT-12 data and the results compared for the 198 samples that had both RNA-seq and Illumina HT-12 data (Fig. 3b). There was a high level of correlation in cycle stage determination using data from the 2 different gene expression platforms, with slightly more variation being seen in the mid-proliferative phase. Additionally, validation using unsupervised methods with initial groupings based on a PCA plot (Supplementary Fig 1) and not using pathology dating information at all also yielded a high level of concordance between the

molecular staging model and the validation model (Supplementary Fig 2). The correlation between the two models was 0.989 and the mean absolute difference between estimated sample cycle times was 1.67%. Peripheral blood estradiol and progesterone levels were not used to help determine cycle stage and could therefore be considered as an independent variable. Estradiol ($N = 159$) and progesterone ($N = 187$) data were plotted against molecular staging model cycle stage and showed typical expected menstrual cycle distributions (Fig. 3c, d).

### Reanalysis of published data
The molecular staging model was used to re-analyse 3 published endometrial gene expression datasets available on GEO (GSE65099, endometrial samples from GSE141549, and GSE180485). These datasets were chosen because they contained endometrium from natural cycles with attached estimates of cycle stage and plots. We first produced a principal component analysis (PCA) plot using our own RNA-seq dataset ($N = 266$) with cycle stage as determined by the molecular staging model (Fig. 4a). This PCA plot has a characteristic pattern with samples clustering according to cycle stage as determined using the molecular staging model, with no outliers. We then generated PCA plots from each of the 3 published datasets. The PCA plot using data from GSE141549[20] is shown in Fig. 4b, with samples labelled as per information in GEO as menstrual, proliferative, secretory and unknown. There is mixing of proliferative samples with menstrual and secretory ones within the PCA plot when using the GEO assigned labels. The same PCA plot using data from GSE141549 but with cycle stage assigned by our molecular staging model has minimal overlap between different cycle stages (Fig. 4c), demonstrating that the molecular staging model accurately aligns with PCA analyses of endometrial gene expression data

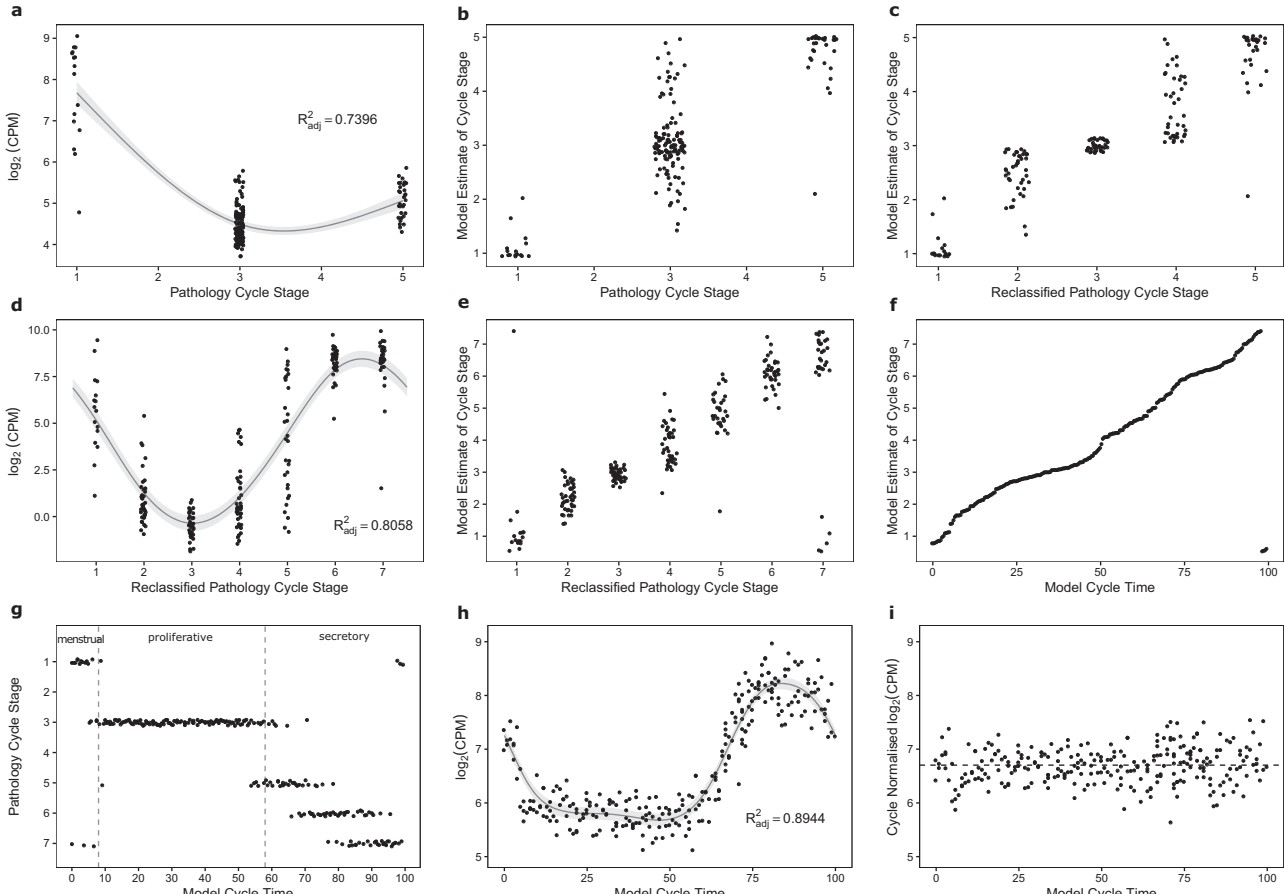

**Fig. 2 | Analysis 2: Development of the molecular staging model to assign cycle stage for the whole menstrual cycle.** Figure 2a. Example of a spline curve fitted to menstrual, combined proliferative and early secretory expression data. The error bands surrounding the regression line represent the 95% confidence interval. Figure 2b. Menstrual, proliferative and early secretory samples with estimated molecular cycle stage calculated from minimum mean squared error data. Figure 2c. Combined proliferative samples reassigned into early, mid and late proliferative groups containing equal numbers. Figure 2d. Example of a spline curve fitted to data from all 7 stages of the cycle using reclassified proliferative cycle stage information and pathology-derived menstrual and secretory staging. The error bands surrounding the regression line represent the 95% confidence interval. Figure 2e. Comparison of cycle time within the 7 cycle stages estimated from the model and pathology-derived cycle stage. Figure 2f. Under the assumption that 236 women underwent surgery at random stages of the menstrual cycle, data from 236 samples were transformed to be uniformly spaced along the x-axis on a 0–100 scale. This transformation allows each sample to be identified as being a percentage of the way through the menstrual cycle. Figure 2g. Plot showing relationship between pathology staging and the percentage of cycle from the molecular staging model. Menstrual is 0–8%, proliferative is 8–58% and secretory is 58–100% of the molecular staging model cycle respectively. Figure 2h, i. Expression data for ENSG00000187231 replotted using derived 'percentage' cycle times and then normalised across the menstrual cycle. The error bands surrounding the regression line represent the 95% confidence interval. Source data are provided as a Source Data file.

across the menstrual cycle. In a similar fashion, but with a smaller dataset from GSE65099[21], samples reported as LH + 6 to LH + 10 do not group in a consistent fashion by PCA (Fig. 4d). When the same samples are assigned cycle stage times by the molecular staging model, the same PCA analysis shows consistent grouping according to cycle stage for all samples, with 2 outlying samples on the PCA plot being reassigned as proliferative and not secretory (Fig. 4e). We repeated the PCA comparison approach using 36 RNA-seq endometrial data sets deposited in GSE180485 (Supplementary Fig 3). These samples are from a study called EndoTime to determine whether the accuracy of timing of luteal phase endometrial biopsies based on urinary ovulation testing could be improved by measuring the expression of a small number of genes using a continuous, non-categorical modelling approach[17]. We produced identical PCA plots with the first showing samples labelled with percentage cycle times derived from our molecular model, and the second using the EndoTime model eLH+ estimates. While our percentage data shows a steady progression through the cycle in concordance with the PCA plot, the eLH+ days 4-5 data points are widely spread (i.e., these samples have significantly different molecular profiles despite the

EndoTime model calling them as similar), and several of the eLH+ days 8–10 samples are grouped together suggesting that this version of the EndoTime model cannot reliably discriminate between these post LH surge days.

## Changes in Endometrial Gene Expression with Increasing Age and Different Ancestries

Using our RNA-seq data ($n = 266$, 20,067 genes analysed) with menstrual cycle staging calculated using the molecular staging model, a total of 60 endometrial genes showed significant changes in expression with increasing age. Examples of 2 significant genes are shown in Fig. 5a. Re-running the age analysis using the original 7 cycle stage pathology data instead of the staging from the molecular staging model reduced the number of age-related significant genes from 60 to 32, providing evidence that the molecular staging model provides a superior approach for identifying differentially expressed genes. To further explore the effects of aging on endometrial gene expression, an additional $n = 87$ Illumina microarray endometrial samples from GSE141549 were analysed and combined with our RNA-seq differential expression results as part of a meta-

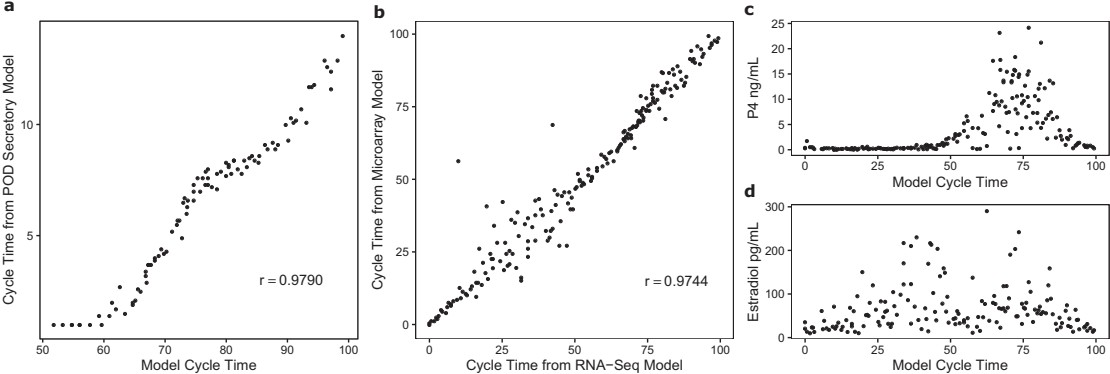

**Fig. 3 | Validation of the NGS Molecular Staging Model.** Figure 3a. For endometrial secretory samples, POD predicted from the secretory molecular staging model was compared against the full molecular staging model. Note, POD 1 is approximately 58% of the way through the cycle. Figure 3b. Illumina HT-12 microarray data were also available from 198 endometrial samples used to generate the molecular staging model. A validation study was run comparing NGS vs Illumina data. The molecular staging model results showed a strong correlation when comparing the 2 different gene expression platforms. Figure 3c, d. Peripheral blood progesterone ($n = 187$) and estrogen ($n = 159$) data plotted against the molecular staging model cycle stage showed expected typical menstrual cycle-stage distribution. Source data are provided as a Source Data file.

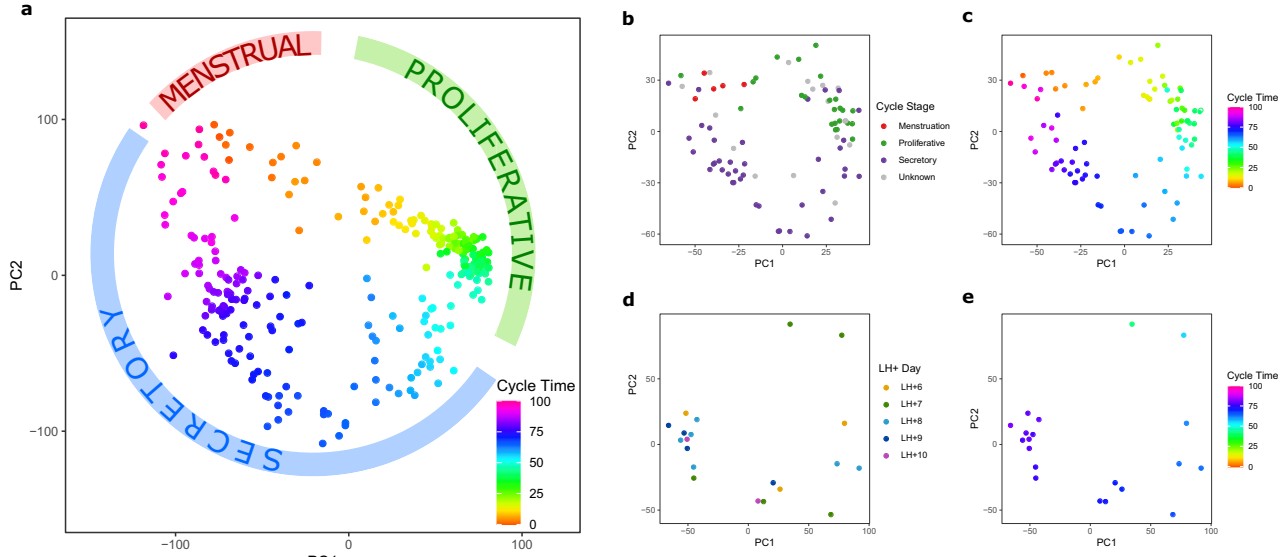

**Fig. 4 | Reanalysis of published data.** Figure 4a. PCA plot using 266 endometrial samples coloured to show the molecular staging model cycle stage. The PCA plot has a characteristic pattern with the samples in cyclic pattern corresponding to approximate cycle stage order. Figure 4b. PCA plot using Illumina microarray data from GSE141549[20], with samples identified as menstrual, proliferative, or secretory by the authors. Note significant mixing of proliferative samples with menstrual and secretory ones. Figure 4c. The same PCA plot using data from GSE141549 with cycle stage assigned by the molecular staging model. Note minimal overlap between different molecular staging model cycle stages. Figure 4d. PCA plot using RNA-seq data from GSE65099[21] with samples identified as LH + 6 to LH + 10 as reported in the study. Figure 4e. The same PCA plot using data from GSE65099 with cycle stage assigned by the molecular staging model. Note reassignment of 2 outlying samples on the PCA plot as proliferative and not secretory. Source data are provided as a Source Data file.

analysis. Considering only genes in both datasets, this reduced the number of genes tested to 12,868, which still included 32 of the 60 significant genes from our original dataset. 65 significant genes were found in the GSE141549 data when analysed on its own. However, when the 2 data sets were combined ($n = 353$), 206 significant genes were identified across the whole menstrual cycle (Supplementary Table 3). We then split the samples into 3 cycle stages; menstrual, proliferative and secretory (equivalent to 0–8%, 8–58% and 58–100% of the molecular staging model cycle respectively) and analysed each stage of the cycle separately (Fig. 5b). Of note, nearly all (218/222 [98%]) of the genes showing significant changes with age were found in samples taken in the secretory phase of the cycle (Supplementary Data 1). A gene ontology enrichment analysis

was run using the 218 genes from secretory samples that changed significantly with age (Supplementary Data 2). The top biological processes enriched with upregulated genes were related to axonemes, cilia and microtubules while the top processes enriched with downregulated genes were related to blood vessels, endothelial cells and angiogenesis.

Ancestry of subjects as defined by a previous study[19] was used to look for differences in endometrial gene expression using pairwise comparisons of each ancestry group. In our Australian population the majority of subjects were of European ancestry, however, despite small numbers in other groups significant differences in gene expression were identified between the groups (Supplementary Fig 4, Supplementary Table 4 & Supplementary Data 3).

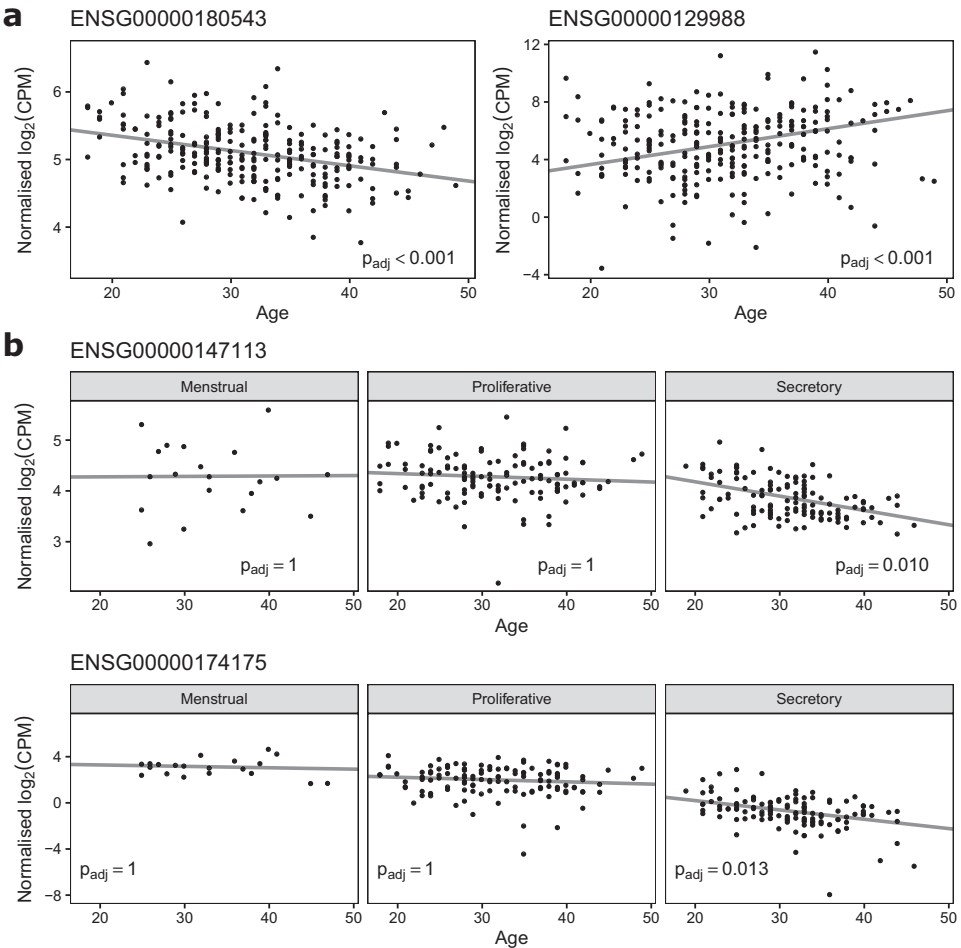

**Fig. 5 | The impact of age on endometrial gene expression.** Figure 5a. Two examples from the 60 endometrial genes that change expression significantly with increasing age. Statistical analysis used two-sided empirical Bayes moderated t-tests implemented in limma[35] with corrections for multiple comparisons performed using the Benjamini-Hochberg method (ENSG00000180543: t = 6.025, $p$ = 5.82e-9, $p_{adj}$ = 5.84e-5; ENSG00000164778: t = −5.886, $p$ = 1.29e-8, $p_{adj}$ = 8.22e-5). Expression data were plotted following normalisation for changing expression across the menstrual cycle ($n$ = 266 RNA-seq samples, 87 samples from GSE141549[20]). Figure 5b. Expression data for 2 genes plotted separately for menstrual, proliferative, and secretory samples ($n$ = 266). Statistical analysis used two-sided empirical Bayes moderated t-tests implemented in limma with corrections for multiple comparisons performed using the Benjamini-Hochberg method (secretory phase ENSG00000147113: t = −5.074, $p$ = 1.43e-6, $p_{adj}$ = 0.010; secretory phase ENSG00000174175: t = −4.429, $p$ = 2.09e-5, $p_{adj}$ = 0.013). Source data are provided as a Source Data file.

## Differential gene expression across the cycle using the molecular staging model

To investigate changing gene expression across the cycle, all samples with RNA-seq data from Study 1 ($n$ = 236) were ranked in chronological order from start to end of the molecular staging model cycle. A 'sliding window' approach was then used to compare differential gene expression (DGE) between samples 1–8 versus samples 9–16, followed by samples 2–9 versus 10–17, then 3–10 vs 11–18 and so on for all samples across the menstrual cycle. Group sizes were arbitrarily set at 8 because this represents 3.4% of the 236 samples or approximately 1 day assuming a mean cycle length of 28 days. Moderated t-tests were used to identify differentially expressed genes with $P$ < 0.05 following multiple testing correction, at each window. Using adjusted P values, 488 unique genes significantly changed expression during menstruation, 44 during the proliferative phase, and 2921 during the secretory phase. Peak times of rapid change in gene expression occurred during menstruation (3% of the way through the cycle), late proliferative (51%), POD3 (66%), POD5 (71%), POD11 (94%) and POD13 (98%) (Fig. 6a). Examples of 12 endometrial genes showing significant and very rapid changes in expression across different stages of the menstrual cycle are provided in Fig. 6b.

The original Endometrial Receptivity Analysis (ERA) publication identified 238 genes that show major changes in expression before, during and after the time of embryo implantation at POD 3–7[22]. Of these 238 genes, 207 were recognised in our NGS data, and 70% of these (145/207) changed expression significantly between cycle times 66 ± 2% and 76 ± 2% (POD 3-7). Supplementary Fig 5 shows the 6 most significantly down-regulated genes and the 6 most significantly up-regulated ERA genes that we identified.

## Discussion

We have developed and validated a method for accurately determining endometrial cycle stage based on global gene expression. We did this by generating mathematically defined curves fitted to RNA-seq expression data from 236 endometrial samples for each of 20,067 genes. From these curves we then found the time of the cycle that minimised a loss function (mean squared error) which gives the best fit for all genes simultaneously for any individual endometrial sample. By placing the 236 samples in chronological order from the start to end of the menstrual cycle and converting the x-axis to percentage, we created a method for both defining how far through the cycle any given sample is, as well as being able to normalise gene expression for cycle

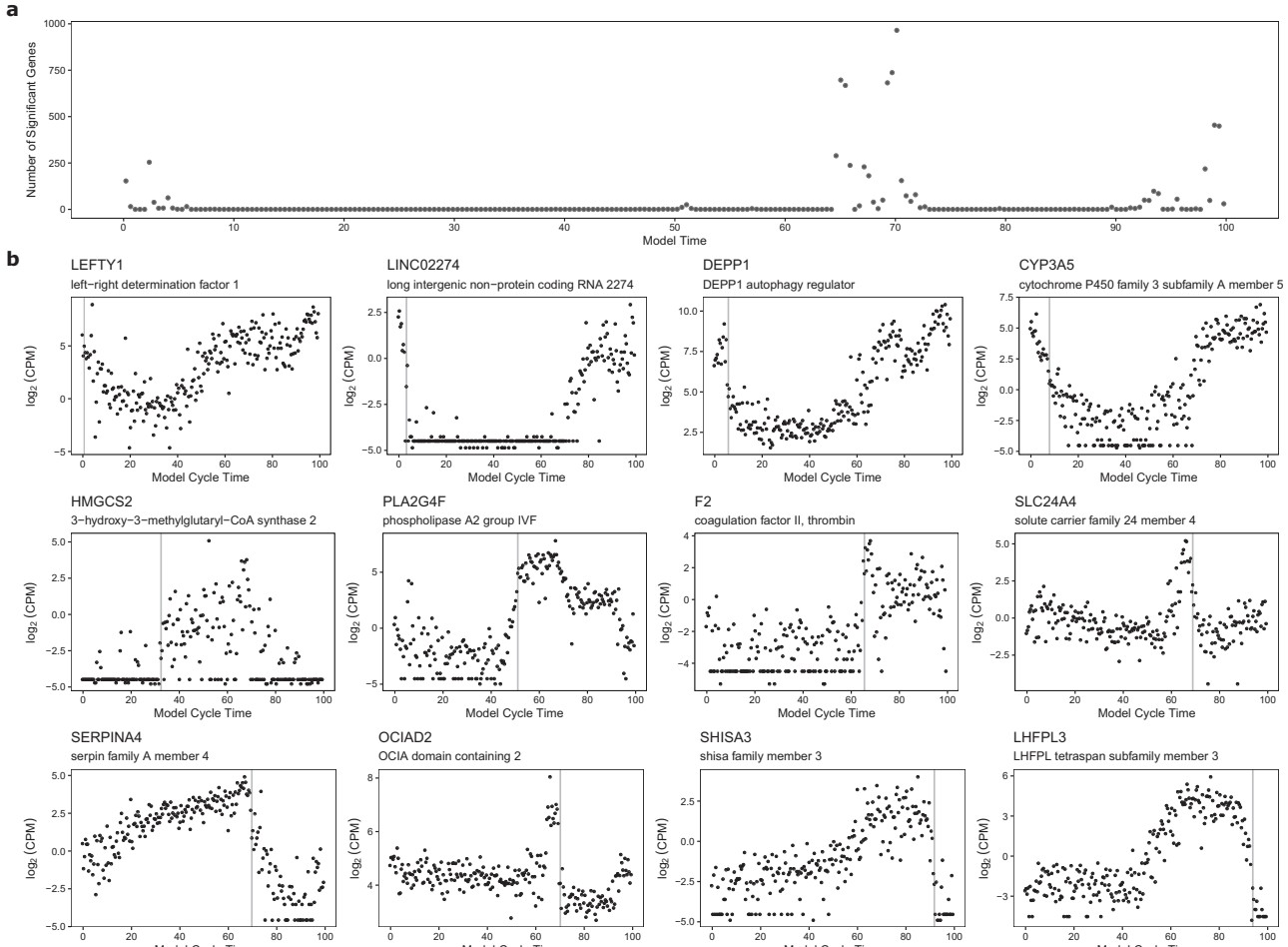

**Fig. 6 | Changing gene expression across the cycle.** Figure 6a. Genes from RNA-seq analysis that significantly change expression ($P_{adj}$ <0.05) over 3.4% of the cycle (approximately equal to a 24-hour window) at different stages of the menstrual cycle. Statistical analysis used two-sided empirical Bayes moderated t-tests, tested relative to a threshold fold change of 1.2, implemented in limma[35] with corrections for multiple comparisons performed using the Benjamini-Hochberg method. Using adjusted *P* values, 488 unique genes significantly change expression during menstruation, 44 during the proliferative phase, and 2921 during the secretory phase. Peak times of rapid change in gene expression approximately correspond to menstrual (3% of the way through the cycle), late proliferative (51%), POD3 (66%), POD5 (71%), POD11 (94%) and POD13 (98%). Figure 6b. Examples of 12 genes that change expression significantly at different times across the cycle. Source data are provided as a Source Data file.

stage so that gene expression differences between any 2 samples can be compared.

Our work has immediate translational application. Being able to accurately identify differentially expressed genes from relatively small numbers of samples will substantially increase the prospect of identifying genes linked to different endometrial pathologies, thus helping to identify biomarkers for diagnosis and potential targets for therapy. Similarly, detailed knowledge of normal gene expression profiles throughout the endometrial cycle will assist in identifying 'off-target' effects during clinical trials of new therapeutics. Accurate cycle staging is also critical for synchronisation of frozen-thaw embryo replacement during in vitro fertilization treatment. Longer-term, improved understanding of endometrial disorders such as heavy menstrual bleeding, endometriosis and adenomyosis will lead to improved outcomes for many women.

Our molecular model for staging the endometrium creates many new research opportunities in benign gynaecology. There is a compelling argument for more research in this field given the considerable financial and quality of life burdens that endometrial-related disorders have on patients and their families. The total direct and indirect costs of abnormal uterine bleeding in the UK and USA were reported more than 10 years ago as greater than £1.2 billion and $37 billion,

respectively[1]. More recently in the USA, heavy menstrual bleeding (HMB) with uterine fibroids had a mean all-cause total cost of $16,762 per patient, while HMB only was $11,135 compared to a control cohort at $6,691[23]. In 2011, in the UK, it was estimated that in England and Wales approximately 1.5 million women were affected by HMB annually. HMB is the fourth most common reason for referral to gynaecology services in the UK and approximately 20% of 1.2 million referrals to specialist gynaecology services concern women with HMB in the UK[1]. In addition to the financial impact, women with menstruation-related symptoms have lower scores on several domains of quality of life such as general health and physical, mental, social and occupational functioning during their periods[24]. A recent review of the economic burden of endometriosis determined the overall direct medical cost range to be from US$1459 to US$20,239 per patient per year, and the indirect costs between US$4572 and US$14,079[2].

Endometrial cycle stage classification is currently based on histopathological criteria first published over 70 years ago[14]. The ongoing need for endometrial cycle staging is evidenced by the fact that this 1950 paper has received over 3,500 citations[25], the most for any study in obstetrics and gynaecology. Accuracy of pathology dating has well established limitations. It has been reported that accuracy in dating the same POD between different pathologists using Noyes criteria was

poor, ranging from 18 to 40%, rising to 60–68% for POD dating within 1 day and to 65–81% for dating within 2 days[15]. The same study reported that accuracy and interobserver reproducibility were unaffected by refresher training, suggesting the limits of histological dating have been reached. By contrast, our model reliably identifies significant differences in gene expression from day to day through the cycle (see Fig. 6) and can assign a time point on a scale more refined than a single day. In addition, our model accurately determines endometrial stage regardless of cycle length and does not rely on measurement of external events such as changes in circulating hormones or fixed time points such as ultrasound detection of ovulation or first day of menstrual bleeding. How accurate these results are in terms of cycle stage is more difficult to ascertain. The comparison of NGS versus Illumina data from the same samples (Fig. 3b) shows that at times of the cycle where gene expression is changing rapidly, such as menstruation and implantation (see Fig. 6a), there is very high concordance between the 2 different methods. This agreement suggests that results at these stages of the cycle are accurate to within less than 3-4% or approximately 1 day. At other times, such as during most of the proliferative phase, there is less agreement, suggesting an accuracy of 7–11% or 2-3 days. By contrast, pathology dating during the proliferative phase is limited to early, mid or late. The use of percentage to describe distance through the cycle removes difficulties in comparing different length cycles. The human menstrual cycle is inherently variable in overall length and in the relative lengths of the menstrual, proliferative and secretory phases[10–13]. This variability makes accurate comparisons between endometrial samples challenging. For example, 10 days post last menstrual period (LMP) can be proliferative endometrium in one woman and secretory in another. Similarly, POD6 can be pre-receptive in one case and post-receptive in another. By defining cycle stage in terms of the percentage of the way through the menstrual cycle that the endometrium is (which still allows precise mapping to events such as menstruation and ovulation), it is possible to accurately synchronise all endometrial samples to their own well-defined point in the cycle. From our data, menstrual is 0–8% of the cycle, proliferative is 8–58% and secretory is 58–100%.

Another advantage of the molecular staging model is that it generates mathematically defined curves (splines) for expression levels of all genes across the cycle. This allows gene expression from different stages of the cycle to be readily normalised for comparison studies. It also provides information on expression profiles for every gene and can be used across different gene array platforms. As part of the validation studies undertaken for the molecular staging model, we used the same methods to derive a similar staging model using Illumina HT-12 data from 198 samples that also had RNA-seq data. This exercise demonstrated a high degree of agreement in the two models' cycle stage between the RNA-seq and Illumina HT-12 data sets, confirming that the molecular staging model method gives robust outcomes when used with different transcriptomics platforms.

Endometrial expression data in GEO and other databases can now be precisely staged and reinterpreted using this methodology. To support this claim, we have reinterpreted 3 published data sets and generated PCA plots to demonstrate that the molecular model provides accurate timing for all samples in each data set. We show good agreement between the molecular staging model and most of the published staging but identify several samples in each published data set that are misclassified. With relatively small sample sizes, a few misclassified samples can severely reduce the ability to identify differentially expressed genes. Based on these results, we suggest the need for caution in interpreting endometrial gene expression studies with small sample sizes and/or where the authors have not been able to incorporate accurate cycle staging into their analyses.

This day-to-day volatility in gene expression during the secretory phase provides a compelling explanation for the consistent lack of agreement between studies comparing gene expression where comparison groups contain all stages of the secretory phase combined. Use of the molecular staging model to accurately assign cycle stage and account for these rapid changes in gene expression data will significantly enhance our ability to compare endometrial gene expression between experimental groups and identify real differences due to phenotype or treatment.

The molecular staging model reveals remarkably synchronised and dramatic changes in expression for many genes at different stages of the cycle. The model delivers highly accurate cycle staging because most of the variance in endometrial gene expression is due to cycle stage, and not other factors. Over 50% of the gene models had an adjusted R-squared value greater than 0.19 and 30% were over 0.36, indicating that the proportion of variance in gene expression explained by modelling cycle time is considerable. Based on this variance, we have been able to identify 3453 genes that change expression significantly within a 3.4% window of the endometrial cycle (which approximately equates to a 24-hour period in a 28-day cycle). Of these, 488 were during menstruation, 44 during the proliferative phase, and 2921 during the secretory phase. These changes are remarkably consistent between samples and synchronised between the 3453 genes within each sample (see for example Fig. 6 and Supplementary Fig 5). It is these cycle-driven changes that provide the statistical strength and remarkable chronological consistency of our modelling across the cycle. If very few genes changed expression across the cycle, most of the expression curves would be relatively flat and there would be no obvious global minimum in the mean squared error plots. Instead, the highly dynamic patterns of gene expression ensure that the mean squared error plots give robust results for predicting a time in the cycle. It is also important to reiterate that the mean squared error solution is simultaneous for all 20,067 genes, giving confidence that significant changes in expression for individual genes are real.

Because most of the variance in endometrial gene expression is linked to cycle stage, cycle stage determination using the molecular staging model is unlikely to be influenced by other variables that impact on gene expression during the menstrual cycle. This is because even if a pathological condition changed the expression 100's of genes, it will only minimally influence the accuracy of the model which is based on significant changes in the expression of several 1000's of genes. The overwhelming contribution of cycle stage variance to the accuracy of the molecular model provides the rationale for the generalisability of the model to endometrial samples taken from women with a wide range of uterine phenotypes and pathologies during the menstrual cycle.

The Endometrial Receptivity Analysis (ERA) is a clinical test used to determine if the endometrium is at the optimal stage of the menstrual cycle for embryo implantation. The test was originally based on a group of 238 genes that change expression rapidly around the time of implantation[22], thus allowing the relative expression profiles to be equated with 'pre-receptive', 'receptive' and 'post-receptive' endometrium. We correctly predicted that using the molecular staging model to identify endometrial genes with rapidly changing expression profiles would also identify many of the ERA genes. As well as identifying 70% of the ERA genes we also found many hundreds of additional genes that change expression significantly around the time of implantation, reinforcing how complex endometrial function is at this critical stage of the menstrual cycle.

Using the molecular staging model to normalise gene expression across the cycle we were able to identify, for the first time, numerous endometrial genes that change expression significantly with increasing age (Supplementary Table 3). Importantly, when we performed a meta-analysis by adding a published dataset from another laboratory and analysing it using our method, the number of significant genes was increased; a result that supports our original finding. An interesting finding from this study was that when examining menstrual, proliferative and secretory stages separately, nearly all the genes that

showed DGE with increasing age did so in endometrial samples taken in the secretory phase. This suggests that it is the endometrial response to progesterone that changes with increasing age, or at least genes with biological actions related to secretory phase events such as implantation, early pregnancy or the late secretory transition to menstruation. One explanation for not identifying significant changes in gene expression during the menstrual phase is low subject numbers (total $N = 28$ with 0 genes significant). But for the proliferative phase from $N = 165$ subjects we found only 4 significant genes while for the secretory phase with $N = 160$ subjects there were 218 significant genes. These data highlight the fundamental difference between proliferative and secretory endometrium and bring in to question the biological relevance of comparing these two very different tissues.

The most significant up and down regulated GO pathways with increasing age have strong biological plausibility (Supplementary Data 2). Upregulated pathways include axoneme assembly, microtubule bundle formation and cilium assembly. Microtubules are present in nearly all cells and play a prominent role, along with ciliated cells, in well characterised changes to the endometrial luminal epithelium during the secretory phase. Although the functional relevance is not clear, the transition from luminal microvilli to pinopodes around the expected time of embryo implantation is thought to be a key morphological marker for endometrial receptivity[26]. More recent evidence has found an increased incidence of luminal epithelial microtubule abnormalities in the endometrium of women with adenomyosis, possibly due to increased inflammation[27]. Blood vessel development and regulation of endothelial cell proliferation were the top 2 down regulated GO biological pathways. The endometrium is a site of rapid cyclical angiogenesis, vascular development and breakdown, with spiral arterioles developing their classical shape and other endometrial vessels growing to match the increasing thickness of the endometrium[28]. Age-related alterations in endometrial angiogenic or vascular potential could be expected to have profound effects on endometrial function, and in particular fertility and subsequent placentation.

Future endometrial DGE studies should account for age when performing statistical modelling. It seems likely that use of the molecular staging model will identify other gynaecological conditions that influence endometrial DGE and hence need to be accounted for in subsequent analyses.

There are limitations to the current study that need to be considered. The 236 endometrial samples used to develop the menstrual cycle staging model in analysis 2 were nearly all taken from women undergoing diagnostic laparoscopy to investigate pelvic pain and suspected endometriosis (IVF subjects with secretory stage endometrium that were used in analysis 1 were not included in the final model). Of the 236 women, 173 were diagnosed with endometriosis. Whether this population can be considered as having normal endometrium is open for debate. However, as reported in Supplementary Tables 1 & 2, 60 of the 236 women had delivered 1 or more live births, with 38 of these 60 also being diagnosed with endometriosis. The ability to carry a normal pregnancy to a successful outcome is evidence that the endometrium was functionally 'normal' in these women, and suggests that if there are pathological differences in the endometrium of women with endometriosis, they are either not universal, or not severe enough to preclude normal endometrial function. Further, the 140 women who had not been pregnant were on average 6.2 years younger than those that had had a successful pregnancy (29.2 vs 35.4 years of age), suggesting that many had not yet tried to conceive and could well have 'normal' endometrium. While it is possible that gynaecological pathologies that affect endometrial gene expression may exist in some of our 236 subjects, the conclusion from this work, as discussed earlier, is that the predominant driver of changing endometrial gene expression is menstrual cycle stage, and not phenotypic factors. If phenotypes that do influence gene expression are identified in the future, for example age and ethnicity in the current work, then these can be corrected for in the model and this will reduce variability and increase sensitivity of future studies.

The relative lengths of the menstrual, proliferative and secretory phases in the molecular staging model are determined by the distribution of women at each stage of the menstrual cycle. We made the assumption that our population of 236 women were randomly distributed across the cycle although it is possible that this was not the case. Hence some phases in the molecular staging model could be compressed or extended over time compared to the normal population. This does not alter the validity of the model for the relative position of the samples in chronological order across the cycle but does mean that mapping of molecular staging model onto cycle stages may update as larger studies are performed.

In conclusion, we have developed and validated a method for accurately determining endometrial cycle stage based on global gene expression. Our 'molecular staging model' reveals significant and remarkably synchronised daily changes in expression for over 3400 endometrial genes at different stages of the cycle, with most change occurring during the secretory phase. These major day-to-day differences in endometrial gene expression provide a compelling explanation for the failure of studies that lack accurate cycle staging to reach consensus on genes of interest. Our study supports selected previous findings and significantly extends existing data. Using the molecular staging model to normalise expression data we can demonstrate significant changes in endometrial gene expression with increasing age. The molecular staging model provides a wealth of new data on endometrial gene expression and establishes a method for investigating the role of the endometrium in critical biological events such as uterine receptivity for embryo implantation as well as gynaecological pathologies such as endometriosis and menstrual disorders.

## Methods

### Subject recruitment and tissue collection

This work was approved by the Human Research Ethics Committee of the Royal Women's Hospital, Melbourne, Australia (Projects 11–24 and 16–43 for Study 1) and Melbourne IVF (Project 13/17 for Study 2), and all subjects gave written informed consent. Study subjects did not receive compensation for participating. A total of 358 endometrial samples were collected for this study, comprising 264 samples taken at the time of diagnostic surgery from women reporting pelvic pain with suspected endometriosis ('Study 1') and 94 samples from individuals undergoing IVF ('Study 2').

Some of the 358 subjects have had data published as part of previous studies investigating genetic regulation of endometrial gene transcription. Specifically, 123 Illumina Human HT-12 v4.0 samples were included in a 2017 study[18], 229 Illumina Human HT-12 v4.0 samples in a 2018 study[29], and 169 & 206 RNA sequencing samples in studies from 2020[19,30].

Endometrial tissue samples (collected by curette or Pipelle biopsy) were obtained for gene expression analysis, along with blood samples for DNA extraction and hormone assays, patient questionnaires, past and present clinical histories, pathology findings and surgical notes. All subjects were premenopausal and free from hormone treatment at the time of biopsy. Endometrial tissue samples were split and either stored in RNA*later* (Life Technologies, Grand Island, NY, USA) at 4°C before being stored at -80°C for total RNA extraction, or formalin fixed and processed routinely for histological assessment.

### Histological dating of endometrium

All 358 endometrial samples were routinely evaluated[14] by at least one experienced pathologist and allocated to one of seven menstrual cycle stages. Menstrual cycle stage definitions, assuming a standardised 28 day cycle, and numbers of samples in each group were as follows:

Stage 1 = menstrual (n = 18, days 1–4), 2 = early proliferative (n = 5, days 5–7), 3 = mid proliferative (n = 104, days 8–11), 4 = late proliferative (n = 29, days 12–15, includes 'interval'), 5 = early secretory (n = 64, days 16–19, or post ovulation days 2–5), 6 = mid secretory (n = 76, days 20–23, or post ovulation days 6–9), 7 = late secretory (n = 40, days 24–28, or post ovulation days 10–14). Twenty-two biopsies taken by Pipelle did not have adequate tissue for reliable pathology reporting. If the pathology report crossed two definitions (e.g. mid/late secretory), then the later cycle stage was used. The exception to this was in the secretory phase where post-ovulatory day (POD) range crossed 2 cycle stages with the majority of the days in the earlier stage, i.e. POD 4–6 = early secretory. If the pathology report only recorded 'proliferative' or 'secretory', then mid-proliferative or mid-secretory was assigned, resulting in elevated numbers in these 2 cycle stages. A subset of secretory stage samples (n = 164) underwent additional evaluation and were assigned an individual post-ovulatory day by a further 1 or 2 pathologists working independently of each other and the previous assessment. Endometrial samples for which the pathologist reported any abnormalities or evidence of exogenous hormone had already been excluded from the study.

### RNA-seq and gene expression array data preparation

Of the 358 endometrial samples, 290 had Illumina Human HT-12 v4.0 performed and 266 underwent RNA-seq (198 samples had both techniques performed). Total RNA was isolated from endometrial samples using the Allprep DNA/RNA Mini Kit (Qiagen, CA) as per the manufacturer's instructions[29]. Briefly, RNA quality was checked using a Bioanalyzer 2100 (Agilent Technologies, CA) and RNA concentration was measured using a NanoDropND-6000 (Thermo Fisher Scientific, USA). All samples were high quality with an RNA integrity number greater than 8. Expression profiles in endometrial tissue were generated by hybridizing 750 ng of cRNA to Illumina Human HT-12 v4.0 Beadchips.

RNA samples were treated with Turbo DNA-free kit (Thermo Fisher Scientific, USA) prior to RNA-seq library generation[19]. Stranded RNA-seq libraries were prepared using the Illumina TruSeq Stranded Total RNA Gold protocol which includes ribosomal depletion (Illumina, USA).

### RNA-seq data processing

Raw sequencing reads were quality checked using FastQC v0.11.7 and MultiQC v1.6. Low quality reads and contaminating HiSeq Illumina adapter sequences were trimmed using Trimmomatic v0.36[31]. Trimmed reads were aligned against the human reference genome (Ensembl *Homo sapiens* GRCh38 release 84) using HISAT2 v2.0.5[32]. Transcript assembly was performed using StringTie v1.3.1 and the Ensembl *Homo sapiens* GRCh38 release 91 reference annotation. Reads mapping to each known transcript were quantified in StringTie[33] to generate transcript-, exon- and intron-level expression matrices for each individual. Raw gene count matrices were also produced using a Python script provided by StringTie.

### Normalisation of RNA-Seq and array expression values

Genes expressed at a low level by RNA-seq, i.e. genes with counts per million (CPM) < 0.5 in > 80% of the samples, were removed. Raw gene counts were normalized for composition bias and total raw reads (library size) using the Trimmed Mean of M (TMM) method in the edgeR R package[34]. Normalized counts were converted to CPM and log2 transformed (log2-CPM). Batch effects from sequencing were removed using the ComBat function from the sva R package (Leek et al., 2020). To load and normalise the microarray data, the R packages limma[35] and lumi[36] were used. Background correction and robust spline normalization (RSN) produced logged values of probe intensity and ComBat was used to remove microarray batch effects. For probes to be included in the array analysis, annotation probe

quality was required to be "Good" or "Perfect" and detection p-value < 0.05 in at least 20% of samples.

### Genotyping

For determination of ancestry, DNA samples from each of the 358 individuals were genotyped on HumanCoreExome or Infinium PsychArray chips (Illumina, USA)[19]. Quality control (QC) was performed in PLINK[29]. Following QC, a total of 282,625 SNPs (hg19) were phased using Shapelt V2 and taken forward to imputation using the haplotype reference consortium reference panel (version r1.1 2016) on the Michigan Imputation Server. SNPs with low imputation quality (R² < 0.8), missing rate >5%, minor allele frequency (MAF) < 1×10⁻⁴, and Hardy–Weinberg equilibrium<1×10⁻⁶ after imputation were removed. The remaining SNP positions were lifted over to the Ensembl genome build 38 (GRCh38) using CrossMap v.0.2.8. SNPs failing to lift-over were assigned to their new GRCh38 position manually based on dbSNP151 GRCh38 patch release 7 (GRCh38.p7), leaving 6,230,993 SNPs for further analysis.

### Hormone assays

Estradiol and progesterone concentrations were measured in bloods taken at the time of endometrial sampling. Some of these hormone data have been published previously[37]. An additional 28 bloods were assayed for progesterone (Serum P was tested on the Roche Cobas e601 immunoanalyser, utilising electrochemiluminescence (ECLIA). The lower limit of detection was 0.06 ng/mL. The inter-assay at a target mean of 1.4 ng/mL returned a CV% of 3.7. The intra-assay at a target mean of 1.5 ng/mL returned a CV% of 6.5). This gave a total of 187 progesterone results and 159 estradiol results that could be plotted against the molecular staging model cycle stage.

### Analysis 1

Development of the 'molecular staging model' to assign cycle stage for secretory stage samples only: All secretory stage samples with RNA-seq data from Study 1 and Study 2 where 2 or 3 independent pathology assessments agreed on the post-ovulatory day (POD) to within 2 days (n = 96 of a possible 180 secretory stage samples) were selected for analysis 1. For each gene, batch-corrected expression was fit to POD using a penalised cubic regression spline using 3 knots implemented with the generalised additive model (gam) function from the mgcv R package[38]. Each curve was used to obtain the expected expression value for each gene for any given day. For each sample, an estimated POD was obtained using the day which minimised the mean squared error (MSE) between the observed expression and the expected expression across all genes.

Alternatively, this procedure can be described minimising *d* in the loss function:

$$L(d) = \sum_{g \in G} \left( y_g - f_g(d) \right)^2 \tag{1}$$

where *d* is the POD constrained between 1 and 14 days, *g* is a gene in gene set *G*, $y_g$ is the observed expression of gene *g*, and $f_g(d)$ is the spline function that describes the expected expression of gene g for day d. Additionally, K-fold cross-validation where K = 5 was performed to ensure the model was not overfitting.

To illustrate that using larger, less precise, units of time can be used to estimate cycle time using the same method, an additional model was built using the pathology-assigned 3 secretory cycle stages (i.e. stages 5, 6, and 7 corresponding to early-, mid-, and late-secretory respectively) instead of the pathology-assigned POD (i.e. 1–14 days). Using the RNA-seq batch-corrected expression data, each gene was fit using the same penalised cubic regression spline (k = 3) as a function of stage, and curves were generated for each gene. Cycle time was estimated for each sample by selecting the time point in the stage (from

4.5 to 7.5) that minimised the MSE between the observed expression and the expected expression across all genes. As validation, cycle time estimated from the 3 stages model was compared to the cycle time estimated from the 14-day POD model.

## Analysis 2

Molecular staging model using 7 pathology stages for the whole cycle with RNA-seq and array expression data: The method for developing the POD of cycle prediction model was replicated with some modifications using all samples from Study 1 ($N = 236$ for NGS) classified into 7 cycle stages by histopathology. Because the majority of the proliferative phase samples were not assigned as early, mid or late by the pathologist, we re-assigned all proliferative samples into early, mid or late by fitting a penalised cubic regression spline (k = 3) using menstrual, proliferative, and early secretory gene expression data. A proliferative time point was estimated using the time-point which minimised the mean squared error between the observed expression and the expected expression across all genes. The proliferative samples were then split into equal groups of early, mid, and late using this time point. This approach assumes that patients presenting for surgery in a public hospital system will approximate a uniform distribution across the menstrual cycle, so the number of early, mid, and late proliferative samples will be approximately equal. Once the proliferative samples were assigned to early, mid and late stages, a penalised cyclic cubic regression spline (k = 8) was fit using the 7 stages. Each endometrial sample is then assigned a 'day' from the model using the time which minimises the mean squared error between the observed expression data for all genes and their corresponding gene models. This is equivalent to minimising the loss function in Eq. (1), except now $d$ can represent any timepoint in the cycle. This 'day' is a relative timepoint in the cycle and does not correspond to a real day. Continuing with the assumption that all 236 women were approximately uniformly distributed across the cycle, the data are transformed so that the distance in time between each sample is identical. This process in effect ranks all the samples in order from the start to the end of the cycle, and no longer relies on assigning days from an idealised 28-day cycle. The x-axis was then scored from 0–100 so that the individual scores for each sample represented the percentage of the way through the menstrual cycle that each sample was.

The gene curves were then refitted using the new derived cycle times for each sample with a penalized cyclic cubic regression spline (k = 30). For visualisation purposes, normalisation of gene expression for cycle stage was then derived by subtracting the expected expression from the observed expression (i.e. calculating the residuals) and re-adding the mean.

## Validation of the molecular staging model

Several studies were conducted to assess how endometrial gene expression data that were generated using the molecular staging model performed.

The first validation study as part of Analysis 1 was to compare results from the secretory stage that had daily POD pathology, and hence substantially more accurate cycle stage information, with results that only used 3 pathology stages across the secretory phase. The similar results from the 2 different pathology inputs validated the use of pathology data dividing the endometrium into 7 cycle stages to develop the molecular staging model across the whole menstrual cycle.

The second validation study was to repeat Analyses 1 and 2 using Illumina HT-12 data and then compare results for the 198 out of 358 samples that had both RNA-seq and Illumina HT-12 data.

The third validation study was to repeat Analysis 2 using unsupervised methods (i.e. without the use of pathology cycle stage information). K-means clustering (k = 5) was performed using the first 2 dimensions after dimensionality reduction principal component analysis (PCA) (Supplementary Fig 1). Clusters were ordered by their

distance between each other, and for each gene, expression was fitted using a penalised cyclic cubic regression spline (k = 6) using the ordered cluster number as a proxy for cycle time. Identical to Analysis 2, for each sample, an optimal cycle time was obtained by calculating the time that minimised the MSE, and subsequently all times were transformed to be uniformly distributed on a scale from 0 to 100. Cycle time estimates from the validation model were compared to the Analysis 2 cycle times, however, since time 0 for the validation model was not expected to correspond to the start of menstruation, time was offset by the earliest menstrual sample (identified by Analysis 2) before assessing performance (Supplementary Fig 2).

The fourth validation study was to determine whether independent endocrine data supported the molecular staging model. For a subset of samples where endocrine data were available, peripheral blood estradiol ($n = 159$) and progesterone ($n = 187$) levels were plotted against the cycle stage from the molecular staging model.

The molecular staging model was also used to re-analyse 3 published datasets available in GEO (GEO DataSets ID: GSE65099[21], GSE141549[20] and GSE180485[17]). Principal component analysis (PCA) plots from these data sets were replotted with the unaltered cycle stage from the original publication and our molecular staging model cycle stage for comparison.

## Application of the molecular staging model

The molecular staging model for gene expression across the menstrual cycle was applied to 3 questions: Does (1) age or (2) ancestry have any influence on endometrial gene expression, and (3) at what stages of the cycle does gene expression change most rapidly? Differential expression analysis was performed with the predicted cycle time as an additional factor in the linear model, modelled as a regression spline. Empirical Bayes moderated t-tests, implemented in the R limma package, were used to assess if genes were differentially expressed.

Age of patient at biopsy was analysed as a continuous variable using all subjects with RNA-seq data ($N = 266$). Since differential expression effects due to age were identified, subsequent differential gene expression (DGE) analyses included age as a factor when fitting the linear model. To further explore the effects of age on endometrial gene expression $n = 87$ subjects from GSE141549 were also analysed, and a meta-analysis was performed using the weighted Fisher's method for combining $p$-values implemented in the metapro R package[39], where weights were proportional to each study's sample size. Ensembl ID's from the current data were matched with the Illumina probe ID's from GSE141549 resulting in 12,868 genes in common between the 2 data sets. If multiple probes matched to the same Ensembl ID, the probe with the greatest mean expression was used. Analyses were run for the whole menstrual cycle, and separately for the menstrual, proliferative and secretory phases. After multiple hypothesis correction, genes with false discovery rate (FDR) corrected $P < 0.05$ were considered to be differentially expressed. Gene ontology enrichment analysis was performed using the clusterProfiler R package[40].

Ancestry of subjects as defined by a previous study[19] was used to look for differences in endometrial gene expression using pairwise comparisons of each ancestry group. Genetic ancestry of subjects was assigned using principal component analysis (PCA). Briefly, genotype data from participants was merged with the 1000 Genomes P3v5 reference data using markers common to both cohorts. Population clusters were determined using the first 5 PCs and were annotated according to the five 1000 genomes super populations (European, Eastern Asian, Admixed America, African, Southern Asian).

To investigate changing gene expression across the cycle all samples with RNA-seq data from Study 1 ($n = 236$) were ranked in chronological order from start to end of the cycle. A 'sliding window' approach was then used to compare DGE between samples 1–8 versus samples 9–16, followed by samples 2–9 versus 10–17, then 3–10 vs

11–18 and so on for all samples across the menstrual cycle. Group sizes were arbitrarily set at 8 because this represents 3.4% of the 236 samples or approximately 1 day assuming a mean cycle length of 28 days. Moderated t-tests with a fold-change cut-off of 1.2, as implemented in limma's treat function, were used to identify differentially expressed genes with $P_{adj} < 0.05$ at each window.

To validate a subset of genes we identified as having rapidly changing expression around the time of embryo implantation, we ran a comparison with genes identified in the original Endometrial Receptivity Analysis (ERA) publication[22]. The ERA identified 238 genes that show major changes in expression before, during and after the time of embryo implantation at POD 3–7. We identified all endometrial genes that showed significant changes in expression within any given 24 hr period at the same time of the menstrual cycle to confirm that our list contained a high proportion of the ERA genes.

### Reporting summary

Further information on research design is available in the Nature Portfolio Reporting Summary linked to this article.

## Data availability

The RNA-seq data generated in this study have been deposited in the Gene Expression Omnibus database under accession number GSE234354 and the Illumina HT-12 microarray data under accession number GSE234368. Additional public datasets used are available on GEO with accession numbers GSE65099 (PMID: 26418742), GSE141549 (PMID: 32859947) and GSE180485 (PMID: 35092277). Homo sapiens reference genome GRCh38 and genome annotation were obtained from Ensembl (release 91). Source data are provided with this paper.

## Code availability

The code for our model is available as an R package at https://github.com/jessicachung/endest (DOI: 10.5281/zenodo.8321573) and an R Shiny application is available at https://github.com/jessicachung/endspect. Analysis scripts for this manuscript can be found at https://github.com/jessicachung/endo_model_paper. Endest is published pursuant to the terms located in the applicable repository at github.com which terms permit reproduction, publication and adaptation of endest solely for non-commercial purposes. Publication of endest at github.com is not subject to the publication terms applicable to Nature publications.

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

## Acknowledgements

We acknowledge the work of Research Nurses Ranita Charitra, Tracy Middleton and Irene Bell who recruited all the participants in this study and Leonie Cann who provided laboratory support. Thank you to the surgeons and theatre booking staff at the Royal Women's Hospital and Staff at Melbourne IVF. We also acknowledge and thank the many women who provided tissues and clinical data that were used in this study. Research reported in this publication was supported in part by National Health and Medical Research Council (NHMRC) project grants GNT1012245 and GNT1105321 (PAWR, GWM, JEG, SJH), NHMRC Fellowship GNT1078399 and Investigators Grant GNT1177194 (GWM), NHMRC Medical Postgraduate Scholarship No. 1055814 and RANZCOG Fellows' Clinical Research Scholarship (2013) (WTT), funds from Melbourne IVF, and the Medical Research Future Fund MRF1199715 (PAWR, MH, SJH, JFD).

## Author contributions

WTT, JC and PAWR were responsible for the conception and design of the study. PAWR, WTT, SJH, JFD, MH and JEG acquired the tissues along with clinical and surgical data. HR, SB and VO provided expert pathological assessment of endometrial biopsies. JC and CS provided bioinformatics and statistical input. RK, JNF, SM and GWM undertook all the gene expression assays and initial QC of the expression data. All authors have read, provided input and approved the final manuscript.

## Competing interests

The method described in this study is the subject of a patent filed at the Australian Patent office (21/6/23 (PCT/AU2023/050559).), by the University of Melbourne with WTT, JC and PAWR as inventors. The remaining authors declare no competing interests.
