## [Peer Review File · Nature Communications]

A molecular staging model for accurately dating the endometrial biopsyREVIEWER COMMENTS

Reviewer #1 (Remarks to the Author):

OVERALL

This manuscript reports an extensive analysis of the endometrial transcriptome of the human uterus. The results are noteworthy and provide a novel method to precisely determine endometrial cycle stage based on global gene expression. The study is original and was extended by reanalysis of a variety of endometrial gene expression data on various platforms during the past two decades. Further, the study evaluated effects of age and ancestry on human endometrial gene expression, which is quite novel. This foundational study will provide essential information to advance our understanding of endometrial-related disorders.

MAJOR AND SPECIFIC COMMENTS

The manuscript contains a well-designed and executed study that is appropriately interpreted with careful consideration of limitations in the Discussion. The methodology is sound, and the results support the conclusions.

The only concern is that the authors need to deposit the RNA-seq data in GEO before publications to ensure the work can be reproduced and allow for new studies to be conducted.

Reviewer #2 (Remarks to the Author):

Dear Authors,

Thank you for submitting your study to Nature Communications. I have read your study entitled "A novel molecular staging model for accurately dating the endometrial biopsy" with great interest. There are several major and minor issues that require change, update, or clarification in the manuscript. Below I wrote my reviewing notes in a point-by-point

manner as they appear in the manuscript. I hope that you find them helpful to improve your manuscript.

Yours faithfully,

Title

1. It is concise and reflects the study PICO

Keywords

1. It is concise and highlighted the rationale, goal, methods, and implication of results.
2. More details on the results can be considered to be added rather than listing various diseases at the end.
3. Overall, it is well written in terms of the language. Only “all nearly all” is confusing and shall be updated accordingly.

Abstract

1. Requires language editing
2. (OR=35.11, 95%CI 2.38 – 517,69; p=0.010) here what does “69” imply? If the authors tried to indicate “517.69”, they should use the consistent decimal point as point or comma.

Introduction

- 1- It can be considered to add briefly the physiological stages of endometrial phases, so that experts on non-medical fields can also follow the manuscript better.
- 2- It can be interesting to also provide some background information on endometrium and polycystic ovary syndrome (PCOS) as it is also very common.
- 3- This study covers only the pre-menopausal women (i.e. not peri- or post- menopausal) which shall be defined and mentioned as endometrium is different in different life stages of a woman. Here, it is the “P” from a classical PICO [Patient or Population, Intervention, Comparison(s), and Outcome(s).]
- 4- The last paragraph shall be before the paragraph starting line 76.

Results

1- Some details are missing on the study participants, such as age mean or median based on the normality of the data, their menopausal status, were there any patients with premature ovarian failure or PCOS other relevant diseases, the number of gravida (mean or median based on the normality of the data), the number of parity (mean or median based on the normality of the data), the number of abort (mean or median based on the normality of the data) by also providing spontaneous or elective, history of placenta rest or other placental disorders such as placenta accrete, history of endometrium cancer or endometrial premalignancies... This list can grow so easily by a gynecologist so that the analysis can be more focused and also reach its clinical implications faster. Hence, creating a table including such relevant background information is needed. Otherwise, the model is very likely to score significantly below expectations from this study in the real life.

2- The patients were selected from a diverse pool.

3- GSE65099 and GSE65099 included "mid-luteal endometrial biopsies from infertile women and patients suffering recurrent pregnancy loss." and "healthy and patient endometrium, peritoneum and patient endometriosis lesions" (see NCBI GEO). It would be great if authors explain their rationale for selecting these datasets and also comment on these specific populations and their model.

4- Line 196: "...nearly all (218/222)...". Please, kindly include percentage.

5- The study provides information on the ancestry but I am afraid not to see the actual numbers (n=,% of the total study population) for specific ancestry.

Figures

1- Figure 1 legend included "Panels 1 to 4" however they were not 100% clear on the figure itself.

2- Figure 1 and 2 can be more colorful to ease the follow-up of the content.

3- Unfortunately, Figure 1 is a very low resolution; hence, it is difficult for readers to interpret those parts fully. I kindly suggest adjusting them accordingly.

4- Figure 1: Step 2, there are two dashed lines representing pathology and one solid line for the model. Do you mean pathologists? Also, please label the pathologists as 1 and 2. The model can be favoring towards one of the pathologists.

Discussion

1- The generalizability of the study is on different population is limited and shall be included in the limitations.

References

1- The Nature Communications state that "References should be numbered sequentially first throughout the text..." (see: <https://www.nature.com/ncomms/submit/how-to-submit>). However, the references in the main text and supplementary did not fill in this criterion.

Language Editing

1. There are minor and major punctuation, spelling, and grammar errors. Upon a proof-reading these shall be corrected by the authors.

Reviewer #3 (Remarks to the Author):

In this study, Teh and coworkers have developed and validated a novel method for precisely determining endometrial cycle stage based on global gene expression. A total of 358 endometrial samples were collected for this study, comprising 264 samples taken from women at the time of surgery for suspected endometriosis ('Study 1') and 94 samples from women undergoing IVF ('Study 2'). Endometrial tissue samples were obtained for gene expression analysis, along with blood samples for DNA extraction and hormone assays, patient questionnaires, past and present clinical histories, pathology findings and surgical notes. All subjects were premenopausal and free from hormone treatment at the time of biopsy. All 358 endometrial samples were routinely evaluated by at least one experienced pathologist. Of the 358 endometrial samples, 290 had Illumina Human HT-12 v4.0 performed and 266 underwent RNAseq (198 samples had both techniques performed).

After conventional normalization of RNAseq data, they did some analyses:

1. Development of the 'molecular staging model' to assign cycle stage for secretory stage samples. Splines were fitted to RNA-seq expression data for each of 20,067 genes from 96 endometrial samples where 2 or 3 independent pathology reports agreed to within 2 post-ovulatory days. For each endometrial sample, an estimated post-ovulatory day was obtained using the day which minimised mean squared error between the observed

expression and the expected expression across all genes.

2. Development of molecular staging model using 7 pathology stages for the whole cycle with RNA-seq and array expression data. A regression analysis was done for all 20,067 genes using the 7 stages of the menstrual cycle. Each endometrial sample was then assigned a 'day' using the time which minimised mean squared error between the observed expression data for all genes and their corresponding gene models. Under the assumption that all 236 women were approximately uniformly distributed across the menstrual cycle, the data were transformed so that the distance in time between each sample was identical. Gene curves were then refitted using the newly derived cycle times for each sample with a regression curve.

Various validation studies, but not on another population, were undertaken using the molecular staging model.

The authors concluded that this model would establish a new method for investigating the role of the endometrium in critical biological events.

The aim of the study is important and the strategy interesting.

However, the study has some problems, some of them have been recognized by the authors but they are anyway quite important.

1. Endometrial samples have been collected from endometriosis and infertile patients and they do not represent 'healthy' subjects. Hundreds of genes have been found to be differentially expressed in endometrium of women affected by infertility and/or endometriosis. Although the Authors have recognized this limit in the Discussion, their mitigation of the issue is unclear. This limit could have been solved validating the model in a group of fertile women (or at least not infertile).

2. Along this line, the real validation of the model, on another population, has not been done. Some validations have been performed but they were more validations of the techniques and of the analysis rather than of the model. Healthy subjects should have been involved.

3. The model was developed based on histopathologic findings. In the Introduction the Authors stated that histopathology of endometrium is the most direct measure of endometrial stage and normalcy, although this is a subjective method that can give variable results. On the other hand, most endometrial samples were evaluated by one pathologist and some of them were not correctly evaluated.

4. Another problem which has been recognized by the Authors was the assumption that the population of 236 women were randomly distributed across the cycle. It is possible indeed that women are referred to interventions or pre-IVF visits in specific menstrual cycle phases. Thus, this assumption may not be valid.

Minor issues:

Panels are not indicated in Figure 1.

Figure 4 does not have the letters.

Abstract is too focused on endometrial pathology which is not the focus of the study.

Reviewer #4 (Remarks to the Author):

This is an interesting study that provides a useful and large new data set and relevant insights into how menstrual cycle timing may be inferred from such data. I have no major concerns regarding the soundness of what is presented. I am concerned, however, by the level of advance in terms of new findings.

It is interesting to see some genes changing expression with age and ethnicity. However, these changes appear to occur in a small number of genes, and are changes of a small extent. For example, the gene chosen in Figure 5a for the representation of the age effect changes from about 5.5 log₂(CPM) at the age of about 16 to about 4.8 log₂(CPM) at the age of about 50. This is roughly a two-fold change in absolute expression level over more than three decades of life, and in the presence of substantial noise in expression throughout age groups. This seems to be a very mild change and its significance is unclear to me.

The advance in terms of being able to achieve more accurate timing of biopsies is not clear to me either. It is impressive to see a correlation of 0.93 in the validation, matching predictions based on HTS data against pathology estimates. But at the same time this implies that the new timing method does not deviate much from timings that were already available for long time. So it may not provide much in terms of new/better timing information.

It would be interesting to establish confidence intervals of timing estimates. When reading the title and seeing the large amount of data, I was hoping the paper would elucidate just how precisely the timing can be estimated, and where the limits may be.

It would have been an added plus to make the computational method easily available and applicable as part of the publication.

This recent publication seems much related, if on much less high-throughput data and limited to the luteal phase, but with an available method:

<https://academic.oup.com/humrep/article/37/4/747/6517177?login=false>

REVIEWER COMMENTS

Reviewer #1 (Remarks to the Author):

OVERALL

This manuscript reports an extensive analysis of the endometrial transcriptome of the human uterus. The results are noteworthy and provide a novel method to precisely determine endometrial cycle stage based on global gene expression. The study is original and was extended by reanalysis of a variety of endometrial gene expression data on various platforms during the past two decades. Further, the study evaluated effects of age and ancestry on human endometrial gene expression, which is quite novel. This foundational study will provide essential information to advance our understanding of endometrial-related disorders.

- We thank reviewer #1 for their positive comments.

MAJOR AND SPECIFIC COMMENTS

The manuscript contains a well-designed and executed study that is appropriately interpreted with careful consideration of limitations in the Discussion. The methodology is sound, and the results support the conclusions.

The only concern is that the authors need to deposit the RNA-seq data in GEO before publications to ensure the work can be reproduced and allow for new studies to be conducted.

- All the computational methods and data sets for our model will be made available if/when the paper is accepted for publication.

Reviewer #2 (Remarks to the Author):

Dear Authors,

Thank you for submitting your study to Nature Communications. I have read your study entitled “A novel molecular staging model for accurately dating the endometrial biopsy” with great interest. There are several major and minor issues that require change, update, or clarification in the manuscript. Below I wrote my reviewing notes in a point-by-point manner as they appear in the manuscript. I hope that you find them helpful to improve your manuscript.

Yours faithfully,

Title

1. It is concise and reflects the study PICO [Patient or Population, Intervention, Comparison(s), and Outcome(s).]

Keywords

1. It is concise and highlighted the rationale, goal, methods, and implication of results.
2. More details on the results can be considered to be added rather than listing various diseases at the end.

- We have made some minor changes to the abstract along the lines suggested by R#2.

3. Overall, it is well written in terms of the language. Only “all nearly all” is confusing and shall be updated accordingly.

- Correction made

Abstract

1. Requires language editing

- We have thoroughly checked and edited the whole manuscript.

2. (OR=35.11, 95%CI 2.38 – 517,69; p=0.010) here what does “69” imply? If the authors tried to indicate “517.69”, they should use the consistent decimal point as point or comma.

- We have not been able to locate this statement in our manuscript. If further information can be provided, we will readily correct any errors.

Introduction

1- It can be considered to add briefly the physiological stages of endometrial phases, so that experts on non-medical fields can also follow the manuscript better.

- We have added a sentence to the second paragraph of the Introduction.

2- It can be interesting to also provide some background information on endometrium and polycystic ovary syndrome (PCOS) as it is also very common. and

3- This study covers only the pre-menopausal women (i.e. not peri- or post- menopausal) which shall be defined and mentioned as endometrium is different in different life stages of a woman. Here, it is the “P” from a classical PICO [Patient or Population, Intervention, Comparison(s), and Outcome(s).]

- Response to points 2 and 3 above: The aim of this study was to develop and validate a new method for accurately determining normal menstrual cycle stage based on changing endometrial gene expression. For this reason, only normally cycling premenopausal women were included in the study, while non-cycling postmenopausal women were not (line 654). Similarly, abnormal endometrium, including from women with PCOS was not part of the study. We agree with the reviewer that it will be interesting to study endometrium from common pathologies such as PCOS now that our model has established normal expression profiles for all endometrial genes across the normal menstrual cycle.

4- The last paragraph shall be before the paragraph starting line 76.

- We have rearranged the paragraphs as requested.

Results

1- Some details are missing on the study participants, such as age mean or median based on the normality of the data, their menopausal status, were there any patients with premature ovarian failure or PCOS other relevant diseases, the number of gravida (mean or median based on the normality of the data), the number of parity (mean or median based on the normality of the data), the number of abort (mean or median based on the normality of the data) by also providing spontaneous or elective, history of placenta rest or other placental disorders such as placenta accrete, history of endometrium cancer or endometrial premalignancies... This list can grow so easily by a gynecologist so that the analysis can be more focused and also reach its clinical implications faster. Hence, creating a table including such relevant background information is needed. Otherwise, the model is very likely to score significantly below expectations from this study in the real life.

- An expanded summary of patient demographic and clinical data is now listed in the first 2 paragraphs of the results. We have added Supplementary Tables 1A and 1B with additional clinical data including gravida, parity, miscarriage, elective termination and fertility intentions as requested.
- We excluded from our study patients with overt endometrial pathologies that were identified by our pathologists (PCOS, cancer, premalignancies). As outlined in our response to the PCOS question above, we agree it will be interesting to investigate a range of endometrial pathologies now that we have tools to do so. This work will be the subject of future studies.
- The final Model for the whole menstrual cycle was developed in Analysis 2 using 236 sets of REFSeq data from subjects having diagnostic laparoscopy for pelvic pain. These subjects are a pragmatic selection of real-life reproductive age women with normal menstrual cycles, not

taking exogenous hormones, and with endometrium assessed as normal by a pathologist. 25% of the women had already had successful pregnancies at the time of endometrial biopsy. We believe our Model, which has been developed to show endometrial cycle stage based on gene expression changes across the menstrual cycle, does perform to 'real world' expectations.

2- The patients were selected from a diverse pool.

- We thank the referee for noting this point.

3- GSE65099 and GSE65099 included "mid-luteal endometrial biopsies from infertile women and patients suffering recurrent pregnancy loss." and "healthy and patient endometrium, peritoneum and patient endometriosis lesions" (see NCBI GEO). It would be great if authors explain their rationale for selecting these datasets and also comment on these specific populations and their model.

- There are limited availability of high quality endometrial whole transcriptome data sets in the public domain. We specifically looked for datasets containing normal menstrual cycle samples. We have analysed all the ones we can find using our Molecular staging Model, and after excluding some due to small size, lack of clinical data or issues with quality control and batch effects, have included GSE141549 (Gabriel, Fey et al. 2020), and GSE65099 (Lucas, Dyer et al. 2016) as good examples to show what our Model can achieve to improve cycle stage timing. As part of our revisions to the manuscript, we have added GSE180485 (Lipecki et al. 2022). In all cases, we have been able to show using PCA plots that our revised cycle staging precisely agrees with the PCA plot, whereas the original published cycle stages have outliers that do not. We have revised the paragraph on 'Reanalysis of Published Data' in the Results section of our manuscript to clarify this point.

4- Line 196: "...nearly all (218/222)...". Please, kindly include percentage.

- We have included % as requested.

5- The study provides information on the ancestry but I am afraid not to see the actual numbers (n=,% of the total study population) for specific ancestry.

- The numbers (N=) for ancestry are recorded in the second column of Supplementary Table 5. We have added % to this information.

Figures

1- Figure 1 legend included "Panels 1 to 4" however they were not 100% clear on the figure itself.

- We have revised Figure 1 to include panel numbers.

2- Figure 1 and 2 can be more colorful to ease the follow-up of the content.

- Our attempts to add colour to Figures 1 and 2 did not (in our opinion) appreciably improve readability. We are happy to revisit this issue if the Editor requests.

3- Unfortunately, Figure 1 is a very low resolution; hence, it is difficult for readers to interpret those parts fully. I kindly suggest adjusting them accordingly.

- We have improved the resolution of Figure 1.

4- Figure 1: Step 2, there are two dashed lines representing pathology and one solid line for the model. Do you mean pathologists? Also, please label the pathologists as 1 and 2. The model can be favoring towards one of the pathologists.

- We have relabelled the dashed lines P1 and P2. However, please note that the final model does not involve (or require) any pathologist input. Step 2 in Figure 1 was part of the process of developing the model where we wanted to make sure that the mean square error approach agreed with the best available pathologist evaluations.

Discussion

1- The generalizability of the study is on different population is limited and shall be included in the limitations.

- A response to concerns about generalisability of the study has been provided earlier in comments to the Editor. In summary, we have revised the Results to clarify and add information on the 236 women used to develop the final model (first paragraph of Results and Supplementary Tables 1A and 1B). These 236 women were not infertile and 25% had had a successful pregnancy. We have also added a third analysis of external data which shows the power of our model to accurately determine cycle stage in different data sets from other laboratories (Supplementary Fig 3). We have included 2 new paragraphs in the Discussion on generalisability of our Model to accurately stage endometrial samples taken from all women during the menstrual cycle and revised the limitations paragraph to support earlier comments about generalisability. We believe this extra work and revisions strengthen our manuscript by clarifying the issue of generalisability of our Model.

References

1- The Nature Communications state that "References should be numbered sequentially first throughout the text..." (see: <https://www.nature.com/ncomms/submit/how-to-submit>). However, the references in the main text and supplementary did not fill in this criterion.

- We have formatted the references as required.

Language Editing

1. There are minor and major punctuation, spelling, and grammar errors. Upon a proof-reading these shall be corrected by the authors.

- We have proof-read and corrected punctuation, spelling, and grammar errors

Reviewer #3 (Remarks to the Author):

In this study, Teh and coworkers have developed and validated a novel method for precisely determining endometrial cycle stage based on global gene expression. A total of 358 endometrial samples were collected for this study, comprising 264 samples taken from women at the time of surgery for suspected endometriosis ('Study 1') and 94 samples from women undergoing IVF ('Study 2'). Endometrial tissue samples were obtained for gene expression analysis, along with blood samples for DNA extraction and hormone assays, patient questionnaires, past and present clinical histories, pathology findings and surgical notes. All subjects were premenopausal and free from hormone treatment at the time of biopsy. All 358 endometrial samples were routinely evaluated by at least one experienced pathologist. Of the 358 endometrial samples, 290 had Illumina Human HT-12 v4.0 performed and 266 underwent RNAseq (198 samples had both techniques performed). After conventional normalization of RNAseq data, they did some analyses:

1. Development of the 'molecular staging model' to assign cycle stage for secretory stage samples. Splines were fitted to RNA-seq expression data for each of 20,067 genes from 96 endometrial

samples where 2 or 3 independent pathology reports agreed to within 2 post-ovulatory days. For each endometrial sample, an estimated post-ovulatory day was obtained using the day which minimised mean squared error between the observed expression and the expected expression across all genes.

2. Development of molecular staging model using 7 pathology stages for the whole cycle with RNA-seq and array expression data. A regression analysis was done for all 20,067 genes using the 7 stages of the menstrual cycle. Each endometrial sample was then assigned a 'day' using the time which minimised mean squared error between the observed expression data for all genes and their corresponding gene models. Under the assumption that all 236 women were approximately uniformly distributed across the menstrual cycle, the data were transformed so that the distance in time between each sample was identical. Gene curves were then refitted using the newly derived cycle times for each sample with a regression curve.

Various validation studies, but not on another population, were undertaken using the molecular staging model.

The authors concluded that this model would establish a new method for investigating the role of the endometrium in critical biological events.

The aim of the study is important and the strategy interesting.

However, the study has some problems, some of them have been recognized by the authors but they are anyway quite important.

1. Endometrial samples have been collected from endometriosis and infertile patients and they do not represent 'healthy' subjects.

- As outlined in response to comments above from the Editor and R#2, our final model is based on endometrial biopsies from 236 women undergoing diagnostic laparoscopy for pelvic pain. None of the IVF patients with infertility were included, i.e. the model is not based on women with infertility. Of the 236 RNASeq datasets used in producing the model, 60 (=25%) came from women who had had at least 1 successful pregnancy, which we believe is the strongest evidence currently available of the presence of a functionally 'healthy' endometrium. We conclude that the overwhelming driver of endometrial transcriptomic change is menstrual cycle stage, which dominates differences due to pathologies such as infertility or endometriosis.
- It is problematic proving that a subject has a normal 'healthy' endometrium, particularly if they have never tried to become pregnant. We have selected a population attending hospital for pelvic pain, which is a symptom not directly linked to any endometrial pathology. We have confirmed normal menstrual cycles and normal pathology by experienced pathologists. Our surgeons have also excluded any overt uterine pathology at laparoscopy and during endometrial biopsy. As mentioned above, 25% of these women have had successful pregnancies. In a pragmatic real-world sense, we believe that from an endometrial perspective, these are typical 'normal' subjects.

Hundreds of genes have been found to be differentially expressed in endometrium of women affected by infertility and/or endometriosis.

- While we agree that hundreds of genes have been reported as differentially expressed in endometrium of women affected by infertility and/or endometriosis, we respectfully point out that none of these studies have been independently repeated or validated. Conversely, there are also numerous studies that have not validated any of the previously reported differentially expressed genes in these groups, some of which report alternative genes that add to the dogma that differences do exist. Currently, it is our view that more work is required on comprehensively phenotyped samples with accurate cycle staging, followed by independent validation studies before differential expression of individual genes can be confirmed. Our staging Model is a foundational contribution that will allow this work to proceed. (Our unpublished opinion currently is that differences in endometrial gene

expression due to endometriosis are very subtle and will require larger data sets than ours to confirm with any statistical certainty).

Although the Authors have recognized this limit in the Discussion, their mitigation of the issue is unclear. This limit could have been solved validating the model in a group of fertile women (or at least not infertile).

- As per our response to the Editors comments and R#2, we have revised the first section of the Results to emphasise that our Model was not based on infertile women. We have added Supplementary Tables 1A and 1B with data on women with successful pregnancies and endometriosis. We have also analysed a third set of published data from a different laboratory (Supplementary Fig 3) which confirms the accuracy of the model in another group of women.

2. Along this line, the real validation of the model, on another population, has not been done. Some validations have been performed but they were more validations of the techniques and of the analysis rather than of the model. Healthy subjects should have been involved.

- The primary aim for our Model is to provide a method for accurately determining menstrual cycle stage based on synchronised expression profiles of all endometrial genes. We have validated it on 3 external data sets and in each case our model gives excellent agreement with PCA plots of the same data, and in each case identifies samples that were mis-classified in the original studies.
- We have developed the model on women with normal menstrual cycles and no discernible endometrial pathology as assessed by one or more expert pathologists. 25% of the women had had successful pregnancies and only 12% reported difficulty conceiving, half of whom had subsequently become pregnant (new information in Supplementary Table 1). We acknowledge in the limitation section that many of these women had endometriosis, however, as discussed above, we do not believe this disease has significantly altered the cycle-dependent changes in gene expression that our Model is based on. As reported in Supplementary Table 1, over half of the successful pregnancies were in women with endometriosis, demonstrating the presence of a functionally competent endometrium despite the endometriosis.

3. The model was developed based on histopathologic findings. In the Introduction the Authors stated that histopathology of endometrium is the most direct measure of endometrial stage and normalcy, although this is a subjective method that can give variable results. On the other hand, most endometrial samples were evaluated by one pathologist and some of them were not correctly evaluated.

- While the Model was initially developed based on histopathologic findings (Analysis 1, 96 endometrial samples with 2 or 3 independent pathology reports agreeing to within 2 post-ovulatory days), part of the novelty of what we discovered was that the changing patterns of gene expression across the menstrual cycle were remarkably consistent for all endometrial samples we analysed. These consistent expression curves allowed us to use the minimum mean square error method to find the best cycle stage estimate for all 20,067 genes simultaneously. We were then able to demonstrate that prior knowledge of pathology was not required to achieve this outcome by using groupings taken from a PCA plot of the data (see Supplementary Figure 1, and section 4 of Results). To reiterate, using a PCA plot as a starting point demonstrates that pathology assessment is not required to develop the final model (although it was essential in developing the concept)

4. Another problem which has been recognized by the Authors was the assumption that the population of 236 women were randomly distributed across the cycle. It is possible indeed that

women are referred to interventions or pre-IVF visits in specific menstrual cycle phases. Thus, this assumption may not be valid.

- We believe our assumption of random distribution across the menstrual cycle is valid for 2 reasons. (1) The final Model developed in Analysis 2 only used subjects undergoing diagnostic laparoscopy for pelvic pain and did not include IVF patients, hence there were no visits at specific menstrual cycle stages. The waiting list for diagnostic surgeries in our public hospital setting is approximately 1 year and scheduling is random with no opportunity based on cycle stage. (2) The variability of the normal menstrual cycle, both in length and the timing of events within the cycle (see literature reviewed in paragraph 3 of the Introduction), adds additional randomness to the distribution of endometrial cycle stage among the 236 subjects.
- It is also relevant to note that our Model ranks all the samples in the chronological order in which transcriptomic (or biological) events are occurring in the endometrium. The time in hours and days between these events, and the overall length of the cycle, is not taken into account. This is why we have used percentage of the cycle completed on the x-axis of our plots, because timing is so highly variable from women to women.

Minor issues:

Panels are not indicated in Figure 1.

- We have redrawn Figure 1 with numbered panels as requested.

Figure 4 does not have the letters.

- Figure 4 is labelled 4A-4E.

Abstract is too focused on endometrial pathology which is not the focus of the study.

- We have made minor changes to the abstract to reduce the focus on pathology and potential translational relevance of the Model.

Reviewer #4 (Remarks to the Author):

This is an interesting study that provides a useful and large new data set and relevant insights into how menstrual cycle timing may be inferred from such data. I have no major concerns regarding the soundness of what is presented. I am concerned, however, by the level of advance in terms of new findings.

- We thank reviewer #4 for comments about the soundness of our work. We respectfully disagree about the level of advance, as we consider our endometrial staging model and results to be a significant foundational advance to the field that will allow reinterpretation of much of what has already been published using endometrial transcriptomics, as well as providing a method for substantially more robust endometrial transcriptomic work in the future. A more detailed response to support this view is given below.

It is interesting to see some genes changing expression with age and ethnicity. However, these changes appear to occur in a small number of genes and are changes of a small extent. For example, the gene chosen in Figure 5a for the representation of the age effect changes from about 5.5 log₂(CPM) at the age of about 16 to about 4.8 log₂(CPM) at the age of about 50. This is roughly a two-fold change in absolute expression level over more than three decades of life, and in the presence of substantial noise in expression throughout age groups. This seems to be a very mild change and its significance is unclear to me.

- We agree that the transcriptomic changes with aging are relatively small, however, we have identified 206 genes that show statistically significant changes with increasing age. This is the first time any endometrial transcriptomic changes due to aging have been identified, and we present this information here as a demonstration of the power and sensitivity of our new

method to detect differentially expressed genes. It is well established that the endometrium in older women can carry normal pregnancies to term, especially if donor oocytes or embryos from younger women are used. This would suggest that aging effects are not significant enough in the endometrium of women under the age of 50 years (as recruited to this study), to prevent relatively normal function. Our results are consistent with this observation.

The advance in terms of being able to achieve more accurate timing of biopsies is not clear to me either. It is impressive to see a correlation of 0.93 in the validation, matching predictions based on HTS data against pathology estimates. But at the same time this implies that the new timing method does not deviate much from timings that were already available for long time. So it may not provide much in terms of new/better timing information.

- Accuracy of pathology dating has well established limitations (Duggan et al, 2001. Cited in para 4 of our introduction). These authors state: *“Accuracy in dating the same POD by each study panel member based on usual practice compared with the reference panel was poor, ranging from 18 to 40%. It improved to 60–68% for POD dating within 1 day and to 65–81% for dating within 2 days.”* Also, *“Accuracy and interobserver reproducibility were unaffected by refresher training, suggesting the limits of histological dating have been reached.”*
- By contrast, our model reliably identifies significant differences in gene expression from day to day through the cycle (see Figure 6) and can assign a time point on a scale more refined than a single day.
- The 0.927 correlation between pathology and our initial model (shown in Figure 1) is based on 96 endometrial samples where 2 or 3 independent pathology reports agreed to within 2 post-ovulatory days. These 96 pathology sections were all technically excellent (21 secretory samples where pathology agreement could not be reached were rejected for Analysis 1). The goal of Analysis 1 was to determine whether we could find agreement between pathology under optimal conditions and our Model. Achieving this outcome gave us confidence to continue developing the Model. The final model is independent of pathology, gives a single, unambiguous cycle time, and is substantially more accurate than pathology.
- Accurate timing of biopsies is only part of what our Model achieves. It also generates mathematically defined expression plots (or splines) across the menstrual cycle for every gene in the endometrium. This is a major new resource that will support a wide range of future mechanistic and clinical studies into endometrial function.

It would be interesting to establish confidence intervals of timing estimates. When reading the title and seeing the large amount of data, I was hoping the paper would elucidate just how precisely the timing can be estimated, and where the limits may be.

- We agree that having a confidence interval for timing estimates would be ideal. Nevertheless, we decided against its implementation due to the complexities involved in calculating an accurate measure of likelihood, given that the variance of gene expression for any particular gene cannot be assumed to be constant throughout the menstrual cycle, as shown in the 4 graphs below. Consequently, we have opted to provide the mean squared error for time points throughout the cycle, which users can plot to infer the goodness-of-fit for the estimated time compared to other menstrual cycle times, instead of introducing a confidence interval that would lack methodological rigor due to difficulties in modelling variance.

Examples of heteroskedasticity in gene expression across the menstrual cycle.

It would have been an added plus to make the computational method easily available and applicable as part of the publication.

- The computational method for our model will be made available as an R package as soon as the paper is accepted for publication. <https://github.com/jessicachung/endest/>

This recent publication seems much related, if on much less high-throughput data and limited to the luteal phase, but with an available method:

<https://academic.oup.com/humrep/article/37/4/747/6517177?login=false>

- We thank the reviewer for bringing this paper to our attention. The Lipecki et al paper is similar to our model in measuring time as a continuous rather than categorical variable (although it still links back to days post-LH surge). However, there are substantial differences in concept, method and outcome that make our model considerably more powerful. Differences include, but are not limited to:
 - Our model covers the whole menstrual cycle, not just part of the early/mid luteal phase, and generates mathematically-defined curves (ie: splines) for average gene expression across the cycle for every gene expressed in the endometrium. The Lipecki et al model assumes monotonic (continuously increasing or decreasing shapes) for gene expression plots over time, which does not reflect what actually happens (for example, see plots below).
 - Our model, based on a mean squared error (MSE) approach provides a cycle stage solution that simultaneously uses all available gene data, providing foundational information for all future endometrial transcriptome studies.
 - The Lipecki et al method is linked to a specific time point in the cycle (the LH surge). We have demonstrated that using our mean squared error approach, unsupervised clustering based on PCA plots (shown in Supplementary Figure 1 of our manuscript) with no prior knowledge of cycle stage is all that is required to generate the Model (even though our initial Model was developed from pathology dating of the biopsies).
 - The EndoTime package produced by Lipecki et al does not have functionality to predict new samples using a pre-trained model, although the authors could add this in the future. As it currently exists, the only function available in the EndoTime namespace is the `endtime_train()` function which requires samples to have a LH+ day defined in order to get the estimated LH time.

- Using our Model, it is possible to identify and select subsets of genes that show significant changes in expression at any given stage of the cycle and then use PCR of those genes to accurately stage endometrial samples around that time. This work is not part of the current paper, but as an example using data in Supplementary Fig 4 of our manuscript, we might select LIF, COMP, SOD2 as upregulated genes and HLA-DOB, KCNJ2 and MAP2K6 as down-regulated genes to replicate the Lipecki model for the early/mid luteal phase.
- Using gene expression data generated by our Model, we note that of the 6 genes used in the Lipecki model, 5 increase expression during the luteal phase, and only 1 decreases expression:

- We posit that the Lipecki model would be more robust if 3 genes that significantly increased expression and 3 genes that significantly decreased expression all at the same time had been selected. Our data also indicate that IL2RB is not ideal due to variable expression during the cycle time of interest, and that IGFBP1 changes expression later in the luteal phase than the other 5 genes and so does not contribute substantially to the EndoTime model. In summary, our Model provides foundational data from which PCR tests for any stage of the cycle can be rapidly developed. The EndoTime model is specific to just 1 time in the cycle.
- Using the RNA-seq data deposited in the GEO database (GSE180485) by Lipecki et al, we ran our model to estimate the menstrual cycle time for the 36 available samples. We then produced identical PCA plots with the Lipecki data (shown below and in the new Supplementary Figure 3), with the first showing samples labelled with our cycle time as a % of the way through the cycle, and the second using the Lipecki et al model eLH+ estimates. Note that whereas our data shows a steady progression through the cycle in concordance with the PCA plot, the eLH+ day 4-5 data points are widely spread (ie: these samples have significantly different molecular profiles despite the Lipecki et al model calling them as similar), and several of the eLH+ days 8-10 samples are grouped together showing that the Lipecki et al model cannot reliably discriminate between these post LH surge days.

- We have included this analysis and PCA plots as Supplementary Figure 3 in the manuscript.
- In summary, for the reasons stated above, we believe our Model provides substantially more, and more accurate data, than the EndoTime model published by Lipecki and co-workers.

REVIEWER COMMENTS

Reviewer #1 (Remarks to the Author):

The authors have satisfactorily addressed my initial comments and those of the other reviewers.

Reviewer #2 (Remarks to the Author):

The authors put sufficient efforts to address the points listed in the first draft. However, its clinical implications are still limited.

Reviewer #3 (Remarks to the Author):

The response of the Authors to the issues raised is very detailed and exhaustive. Generalizability of the results on 'healthy' endometrial samples due to the inclusion of different populations in the analysis in terms of 'molecular endometrial pattern' was a problem of the study. In the Results section, the subjects included have been better described and the problem has been extensively discussed in the Discussion.

Reviewer #4 (Remarks to the Author):

1) The finding on gene expression differences is mentioned in the abstract, but is reflected inside the main manuscript merely by the phrase (“... significant differences in gene expression were identified ...”) and a pointer to supplementary material is given. If these findings are deemed of sufficient relevance to be mentioned in the abstract, then these ought to be presented and analysed in the main body of the paper.

2) I still cannot see why the results should be interpreted as the discovery of significant differential gene expression with age. For example, the gene show-cased in Figure 5A, which is listed under its technical name (ENSG00000180543) rather than its natural language-name (“Testis-Specific Y-Encoded-Like Protein 5”), has a log fold-change of 0.029 according Supplementary Table 3. A log fold-change of 0.029 is a fold-change of 1.029425, so a change in gene expression of 3%. Most papers analysing differential gene expression will discard genes that change by less than 100% (fold-change of 2) as biologically insignificant. While the threshold of 2 for the fold-change is by no means inevitable, deviating so much from this practice and interpreting differences as low as 1% (such as, for example, for “Testis Expressed 10”) as significant requires explanation, to say the least. Very few genes show a change of more than 10%, and none have the sort of fold-changes normally considered biologically significant.

In my mind it is surprising that larger changes in gene expression are not found, given the vast range of ages among subjects and the known differences in hormone levels which one might expect to affect the endometrium much more, as a knock-on effect of ovarian ageing onto the endometrium, even if the endometrium itself might age much less. How can it remain so stable while hormone levels are altered? Is there a compensatory mechanism in

place? So the finding may be an unexpected level of stability in gene expression over age, rather than “significant differences”.

In this context, it would be good to know how the subjects were split into groups for the t-tests. I am assuming it must be “young” vs “old”, but how is the age threshold decided? Or should one simply compare the, say, 100 oldest to the 100 youngest instead to get a more clear discrimination between groups?

3) It is not clear to me that there is a methodological advance:

- Research groups who produce transcriptomic data sets with many samples will be able to observe menstrual cycle timing accurately in PC1 and PC2 of a standard PCA – as the authors exemplify themselves in their Supplementary Figure 3. On the other hand, whether research groups with small numbers of transcriptomic data sets can benefit is unclear as batch effect correction may be an issue.

- The comparison to EndoTime is not a fair comparison, as EndoTime was trained on qPCR data and uses only 6 genes, rather than about 20,000 genes measured in RNA-seq. The PCA plots shown in Supplementary Figure 3 are not identical which makes it hard to see whether EndoTime really performs less well for late samples. The proposed method also shows variability in this group of samples and not all dots can be matched by position by eye. If a comparison is to be made, why not train EndoTime on all samples rather than just using this much smaller data set?

4) It is claimed in the reply to reviewers that the model is “substantially more accurate than pathology”. If so, then why is this not better described in the manuscript? For example, if using pathology instead of whole-transcriptome measurements to infer menstrual cycle time, what is extent of improvement? By how many days should one expect pathology estimates to deviate from the truth on average?

Minor:

5) The term “EndoTime” is used in the manuscript without being introduced. It is unlikely that many readers will be familiar with this method so it should be briefly introduced before the term is used.

6) Supplementary Table 3 is stated to list “significant differential expression”, but includes some genes with high adjusted p-values (such as, for example, 0.7841).

7) A number of methodological claims made in the reply to reviewers are wrong:

- if variance is changing over time, then it does make assessment of confidence in estimations harder, but it is still possible. EndoTime tackles this issue as well and derives a confidence score for predictions (though not a confidence interval).

- EndoTime does not depend on marker genes being monotonous, non-monotonous marker genes can be used.

- The EndoTime method could be applied to the entire menstrual cycle, by providing clinical time estimates of a different time domain than LH+.

- It is not true that the predictive value of a timing marker gene differs depending on whether the gene goes up or down over time.

Reviewer #4 (Remarks to the Author):

1) The finding on gene expression differences is mentioned in the abstract, but is reflected inside the main manuscript merely by the phrase (“... significant differences in gene expression were identified ...”) and a pointer to supplementary material is given. If these findings are deemed of sufficient relevance to be mentioned in the abstract, then these ought to be presented and analysed in the main body of the paper.

The main manuscript contains sections in both the Methods and Results under the headings “Application of the molecular staging model” and “Changes in Endometrial Gene Expression with Increasing Age and Different Ancestries” that detail methods and results for the age-related part of this work. Figure 5 provides examples of changing gene expression with age, and there are 3 paragraphs (Nos 12, 13 and 14) in the Discussion covering our findings about DGE with age, including comments in paragraph 13 about GO pathways and biological plausibility of the findings. We then provide a gene list and GO pathways in Supplementary Tables 3 and 4. We have attempted to make this balance between the main manuscript and the Supplementary information appropriate for the age-related results given that they are only 1 part of the second aim of the study (as stated in the last paragraph of the introduction). We would be happy to move more information from the Supplementary material into the main manuscript if the Editor believed this would improve the manuscript.

2) I still cannot see why the results should be interpreted as the discovery of significant differential gene expression with age. For example, the gene show-cased in Figure 5A, which is listed under its technical name (ENSG00000180543) rather than its natural language-name (“Testis-Specific Y-Encoded-Like Protein 5”), has a log fold-change of 0.029 according to Supplementary Table 3. A log fold-change of 0.029 is a fold-change of 1.029425, so a change in gene expression of 3%. Most papers analysing differential gene expression will discard genes that change by less than 100% (fold-change of 2) as biologically insignificant. While the threshold of 2 for the fold-change is by no means inevitable, deviating so much from this practice and interpreting differences as low as 1% (such as, for example, for “Testis Expressed 10”) as significant requires explanation, to say the least. Very few genes show a change of more than 10%, and none have the sort of fold-changes normally considered biologically significant. In my mind it is surprising that larger changes in gene expression are not found, given the vast range of ages among subjects and the known differences in hormone levels which one might expect to affect the endometrium much more, as a knock-on effect of ovarian ageing onto the endometrium, even if the endometrium itself might age much less. How can it remain so stable while hormone levels are altered? Is there a compensatory mechanism in place? So the finding may be an unexpected level of stability in gene expression over age, rather than “significant differences”.

In this context, it would be good to know how the subjects were split into groups for the t-tests. I am assuming it must be “young” vs “old”, but how is the age threshold decided? Or should one simply compare the, say, 100 oldest to the 100 youngest instead to get a more clear discrimination between groups?

Please note that age of patient at biopsy was analysed as a continuous variable (as stated in the Methods), and not dichotomised into groups (ie: young versus old). Using this approach, we identified subtle but statistically significant changes in gene expression with age as described in the Results. The effects of ovarian aging (eg longer and/or irregular cycles) do not directly influence the results as these variables are accounted for by cycle stage normalisation. The

ability to accurately identify cycle stage and then normalise gene expression to account for changes due to cycle stage are key advances provided by our model. We have included changes in gene expression due to aging in the final model.

3) It is not clear to me that there is a methodological advance:

- Research groups who produce transcriptomic data sets with many samples will be able to observe menstrual cycle timing accurately in PC1 and PC2 of a standard PCA – as the authors exemplify themselves in their Supplementary Figure 3. On the other hand, whether research groups with small numbers of transcriptomic data sets can benefit is unclear as batch effect correction may be an issue.

While research groups can already observe the pattern of menstrual timing in a PCA plot of their samples, they are unable to get a numerical value describing how far along the menstrual cycle the sample is from. Using our software package, researchers can obtain an estimate in the form of a numerical value from 0 to 99, which can be used as a covariate in downstream analysis such as differential expression analysis.

Regarding datasets with a small number of samples, as our method assesses each sample independently, the size of the dataset is immaterial. Our software package will return the same estimate of cycle time for a given gene expression profile. This can be attributed to the fact that our package doesn't require to be trained on external data as the model coefficients for gene expression estimates across the cycle are stored in the package.

With regard to batch effects, our method uses quantile normalisation to match each sample to our reference sample, so batches aren't taken into account. If the user chooses to, they have the option to perform batch correction before using our method. However, in our experience, we have found batch effects do not have a large effect when predicting cycle time with our method. While any single gene may have large differences between batches, the cumulative analysis of tens of thousands of genes within our model effectively reduces these effects to unbiased noise.

- The comparison to EndoTime is not a fair comparison, as EndoTime was trained on qPCR data and uses only 6 genes, rather than about 20,000 genes measured in RNA-seq. The PCA plots shown in Supplementary Figure 3 are not identical which makes it hard to see whether EndoTime really performs less well for late samples. The proposed method also shows variability in this group of samples and not all dots can be matched by position by eye. If a comparison is to be made, why not train EndoTime on all samples rather than just using this much smaller data set?

Our previous comments comparing our model to EndoTime were made to distinguish the differences between the two in response to reviewers' questions. We agree that it is not a 'fair' comparison, but this is not meant as a criticism of EndoTime, and we have not included any of these comments in the main manuscript. As outlined in our previous response to the reviewers, we believe our molecular model has major differences to EndoTime and is a significant advance in the field that provides a foundational change to the way we think about the endometrial cycle.

We have also re-generated the PCA plot for Supplementary Figure 3 using the 36 samples from GEO (accession GSE180485) with DESeq2's rlog transformation function to make the plots identical.

4) It is claimed in the reply to reviewers that the model is "substantially more accurate than pathology". If so, then why is this not better described in the manuscript? For example, if using

pathology instead of whole-transcriptome measurements to infer menstrual cycle time, what is extent of improvement? By how many days should one expect pathology estimates to deviate from the truth on average?

The question “By how many days should one expect pathology estimates to deviate from the truth on average?” is highly pertinent and not possible to answer because there is no agreed definition of ‘truth’ for endometrial cycle stage. As discussed in paragraph 4 of the Introduction, most measures of endometrial cycle stage are indirect and/or have inherent inaccuracies. We provide a more detailed discussion on the inaccuracies of pathology dating in paragraph 4 of the Discussion. In response to the reviewers question we have expanded paragraph 4 of the Discussion to include the following comments about the accuracy of our molecular model for dating the endometrium:

“How accurate these results are in terms of cycle stage is more difficult to ascertain. The comparison of NGS versus Illumina data from the same samples (Figure 4C) shows that at times of the cycle where gene expression is changing rapidly, such as menstruation and mid-secretory (see Figure 6a), there is very high concordance between the 2 different methods. This agreement suggests that results at these stages of the cycle are accurate to within less than 3-4% (approximately 1 day). At other times, such as during most of the proliferative phase, there is less agreement, suggesting an accuracy of around 7-11% (2-3 days). By contrast, pathology dating during the proliferative phase is limited to early-, mid- or late-proliferative.”

Minor:

5) The term “EndoTime” is used in the manuscript without being introduced. It is unlikely that many readers will be familiar with this method so it should be briefly introduced before the term is used.

We have added the following to the Reanalysis of Published Data in the Results section of our manuscript.

“These samples are from a study called EndoTime to determine whether the accuracy of timing of luteal phase endometrial biopsies based on urinary ovulation testing could be improved by measuring the expression of a small number of genes using a continuous, non-categorical modelling approach.”

6) Supplementary Table 3 is stated to list “significant differential expression”, but includes some genes with high adjusted p-values (such as, for example, 0.7841).

We thank the reviewer for spotting this error. The data in Supplementary Table 3 was extracted from a larger spreadsheet and the P values selected were from the analysis that only used RNA-seq samples from the current study and not the combined analysis with GSE141549. We have corrected this error.

7) A number of methodological claims made in the reply to reviewers are wrong:

- if variance is changing over time, then it does make assessment of confidence in estimations harder, but it is still possible. EndoTime tackles this issue as well and derives a confidence score for predictions (though not a confidence interval).

We acknowledge that the presence of heteroskedasticity does not preclude the ability to assess confidence, however, our method of implementing this would require a hierarchical Bayesian approach to model variability over time, which would allow for likelihood estimations. This approach is beyond the scope of the present study.

- EndoTime does not depend on marker genes being monotonous, non-monotonous marker genes can be used.

We acknowledge that that EndoTime methodology can be used for any temporal patterns. We do note however that Lipecki et al state that temporal patterns in the current panel genes are limited to monotonic shapes.

- The EndoTime method could be applied to the entire menstrual cycle, by providing clinical time estimates of a different time domain than LH+.

We acknowledge that the EndoTime method can be used more flexibly than just the secretory phase by providing different time values other than LH+ days. Nevertheless, the current implementation of EndoTime in R requires an initial time estimate for input samples (although we acknowledge this could be changed in future versions of the software), whereas our method does not require training the model for new datasets or an initial time estimate for samples.

- It is not true that the predictive value of a timing marker gene differs depending on whether the gene goes up or down over time.

We agree with this comment. What we were trying to point out is that selecting the 6 most suitable genes to run EndoTest for different stages of the cycle is not straightforward. To get the best discrimination over time the ideal selection will include genes that are changing expression rapidly over the time of interest, with some increasing and some decreasing. Conversely, if the 6 selected genes all have identical expression patterns, or do not change expression much over the time of interest, then the predictive value of EndoGene will be reduced.

REVIEWER COMMENTS

Reviewer #4 (Remarks to the Author):

Overall, no substantial changes were made in response to my criticisms, and our dispute is not really advancing further. I think we simply do not agree on a number of key issues. The editor will have to take judgement.

I list the reasons why I do not see my main points as resolved:

- "Please note that age of patient at biopsy was analysed as a continuous variable (as stated in the Methods), and not dichotomised into groups"

"The effects of ovarian aging (eg longer and/or irregular cycles) do not directly influence the results as these variables are accounted for by cycle stage normalisation"

"identified subtle but statistically significant changes"

Statistical significance alone does not mean much, as the effect size can still be very small. Sometimes, the term "substantially significant" is used and contrasted with "statistical significance". The combination of not providing a dichotomised analysis, reporting fold-changes that are much lower than what is normally considered biologically relevant, and the potential of timing identification itself removing some of the age-related effects, makes it impossible for the reader to make up their mind about your data. Is the endometrial transcriptome essentially not ageing, apart from hormone-mediated changes in cycle length? Or may it be ageing a lot, but the continuous analysis does not reveal age-related changes well and therefore undersells these?

- "Using our software package, researchers can obtain an estimate in the form of a numerical value from 0 to 99"

One can order samples using PCA alone. After determining the order of samples, there is little extra information in assigning numbers to each sample.

- It is not rare that for expression data sets from different labs PC1 distinguishes samples by lab rather than a biologically meaningful parameter. So batch effects are a potential issue, even if substantial batch effects are not seen all of the time.

- "The question 'By how many days should one expect pathology estimates to deviate from the truth on average?' is highly pertinent and not possible to answer because there is no

agreed definition of 'truth' for endometrial cycle stage"

You are proposing a computational definition of endometrial cycle stage so you could use your own definition and answer the question regarding confidence intervals of estimates. It is a shame that with an input size of tens of thousands of measured expression levels per sample the only output generated is a single number without a confidence score or confidence interval.

- "Conversely, if the 6 selected genes all have identical expression patterns, or do not change expression much over the time of interest, then the predictive value of EndoGene will be reduced."

Two very different things have been mixed together here:

1) genes that do not change much over time. Of course these will not carry much temporal information.

2) genes that do change readily but have the same temporal pattern. As each is measured independently these do provide independent information on timing, irrespective of similarity in their temporal pattern.

RESPONSES TO REVIEWER COMMENTS (in red)

Reviewer #4 (Remarks to the Author):

Overall, no substantial changes were made in response to my criticisms, and our dispute is not really advancing further.

While accepting that points of difference still exist, we note that all but 1 of the initial issues and most of the issues raised subsequently by Reviewer #4 have been satisfactorily addressed and have resulted in changes that have improved the manuscript.

I think we simply do not agree on a number of key issues. The editor will have to take judgement.

I list the reasons why I do not see my main points as resolved:

- "Please note that age of patient at biopsy was analysed as a continuous variable (as stated in the Methods), and not dichotomised into groups"

"The effects of ovarian aging (eg longer and/or irregular cycles) do not directly influence the results as these variables are accounted for by cycle stage normalisation"

"identified subtle but statistically significant changes"

Statistical significance alone does not mean much, as the effect size can still be very small.

Sometimes, the term "substantially significant" is used and contrasted with "statistical significance".

We have identified over 200 endometrial genes using appropriate and widely accepted statistical methods (listed in Supplementary Table 3) that show changes in expression with age. Regarding biological significance, we show in Supplementary Table 4 the remarkable consistency in the biological processes regulated by these genes, with all 15 significantly upregulated pathways relating to axenome/cilia/microtubules, and at least 20 of the significantly down regulated pathways relating to vasculature/haemostasis. Functional studies based on these data are beyond the scope of the current manuscript, however we consider these results biologically plausible, and they provide a sound basis for ongoing studies into the effects of aging.

The combination of not providing a dichotomised analysis, reporting fold-changes that are much lower than what is normally considered biologically relevant, and the potential of timing identification itself removing some of the age-related effects, makes it impossible for the reader to make up their mind about your data.

Using age as a continuous variable is much more statistically robust than using a dichotomised approach. See Altman et al, BMJ, 2006 and Donner et al, Biometrics, 1994.

<https://www.ncbi.nlm.nih.gov/pmc/articles/PMC1458573/>

<https://www.jstor.org/stable/2533400>

As discussed in these publications, with a dichotomised approach age boundaries and group size are set arbitrarily by the investigator and can be subjectively 'cherry-picked' to maximise differences in differential expression between groups. By contrast, using all the data with age as a continuous variable removes these subjective options from the analysis and is statistically more robust.

The size of the fold-change does not necessarily inform biological relevance. We are analysing whole tissue. There can be significant changes in subsets of 1 cell type within a tissue that may have profound effects on biological function. Our data points to 2 relatively small cell populations: ciliated epithelial cells (as indicated by the GO pathways for axenome/cilia/microtubules), and parts of the vasculature (as indicated by GO pathways for vasculature/haemostasis).

Of note, in 2022 around the time we submitted our manuscript for review, Devesa-Peiro et al published in Human Reproduction <https://academic.oup.com/humrep/article/37/4/762/6516039> analysing transcriptomic data from 27 women (GSE4888 from a 2006 endometrial transcriptomics study <https://www.ncbi.nlm.nih.gov/geo/query/acc.cgi?acc=GSE4888>) with a further validation set of 20 women. In agreement with our findings, they reported endometrial gene expression changes

with age, with the most prominent being up-regulation of ciliary processes, found in both their main and validation datasets. Down regulated pathways included vascular and angiogenesis functions. Unfortunately, we have reservations about the methods Devesa-Peiro et al. employed. In their main analysis of 27 women, all the healthy controls were under 35 years old and all except one of the non-healthy samples were over 35. They performed batch correction for cycle stage and diagnostic groups while retaining age effects, however, since patient age was highly correlated with the healthy controls, this likely introduced false signals in the age effect due to multicollinearity. (See <https://academic.oup.com/biostatistics/article/17/1/29/1744261> for an in-depth explanation). This may help to explain why they found 5778 differentially expressed genes. For this reason, we have not included this reference in the discussion of our manuscript.

Is the endometrial transcriptome essentially not ageing, apart from hormone-mediated changes in cycle length? Or may it be ageing a lot, but the continuous analysis does not reveal age-related changes well and therefore undersells these?

We have demonstrated using a large sample size (N=236) that once the changes due to cycle stage are normalised, there are over 200 endometrial genes that change expression significantly with increasing age, with the majority of changes seen during the secretory phase of the menstrual cycle. As we have noted previously, the endometrium of women up to age 50 is able to support normal pregnancy, although possibly with a higher pregnancy wastage rate. This ongoing functional competence supports the concept that while changes with aging do occur, they are not significant enough to preclude pregnancy.

- "Using our software package, researchers can obtain an estimate in the form of a numerical value from 0 to 99"

One can order samples using PCA alone. After determining the order of samples, there is little extra information in assigning numbers to each sample.

We disagree. A PCA plot shows samples in reduced dimensions based on similarity when transformed with new axes represented by principal components (usually the first 2 components capture most cycle stage effects, however this is not always the case. Lower components can be used and may contain important additional information). Our method assigns samples a time point from 0-100, and while we have used the agreement between our results and PCA plots as one example of validation of our new methodology, it is not possible to obtain cycle staging or ordering from a PCA plot that is as accurate as that provided by our method, especially with batch effects present as discussed below.

- It is not rare that for expression data sets from different labs PC1 distinguishes samples by lab rather than a biologically meaningful parameter. So batch effects are a potential issue, even if substantial batch effects are not seen all of the time.

We agree that it's not rare for PC1 to contain batch effects, however, we disagree that batch effects are an issue for our method. As stated in our previous response to this point, batch effects do not have a large effect when predicting cycle time with our method. As evidence in support of this statement we have performed an additional analysis of 9 expression datasets selected from different laboratories. The full computational workbook for this analysis is available at:

https://rpubs.com/jessicachung/endest_batch_effects (we have also submitted a PDF printout of the computational workbook with this response).

To summarise this analysis, we used raw un-normalised counts from 9 different datasets as the input to estimate a menstrual cycle time for each sample. A combined PCA plot using log-counts-per-million values shows obvious batch effects in the data as predicted by Reviewer #4:

The same batch effects are obvious in the same PCA plot coloured by our estimated model time:

However, when we subset the same log-counts-per-million values to individual PCA plots for each of the 9 studies (so the PCs represent the axes of most variation within the study, not between studies), we get a high degree of agreement with the estimated model time. Some of the studies have LH+ values which we can also display as an additional comparison:

GSE106602

As shown above, batch effects do not influence assignment of cycle stage within individual studies. Cycle stage by our method and by LH generally agree, however, most LH plots have some incorrectly categorised samples. E.g. in the top right cluster of the GSE106602 PCA above, the LH plot has 3 samples labelled as LH+7 clustered with a group of samples that are labelled LH+2. Our method assigns all these samples that cluster together by PCA as around 50% of the way through the cycle.

In conclusion, we stand by our statement that batch effects do not have a large effect when predicting cycle time with our method. (All nine PCA plots are available in the computational workbook at the link above, as well as the submitted PDF, under the "Separate study PCA plots" section.)

- "The question 'By how many days should one expect pathology estimates to deviate from the truth on average?' is highly pertinent and not possible to answer because there is no agreed definition of 'truth' for endometrial cycle stage"

You are proposing a computational definition of endometrial cycle stage so you could use your own definition and answer the question regarding confidence intervals of estimates. It is a shame that with an input size of tens of thousands of measured expression levels per sample the only output generated is a single number without a confidence score or confidence interval.

The original question from Reviewer #4 was 'By how many days should one expect pathology estimates to deviate from the truth on average?' Our answer about the lack of a 'truth' for endometrial cycle stage is unchanged.

The assumption that we should be able to derive confidence intervals for our cycle stage estimate with our method is not correct for technical reasons. Our definition of endometrial cycle stage is based on a best fit solution using mean squared error of expression values of all genes in the endometrium (in contrast to a maximum-likelihood-based approach). In addition, the output of the `estimate_cycle_time` function from our endest package returns three items in an R list: the estimated time (a single number for each input sample), the MSE matrix (a matrix containing the mean squared error for each sample at each time point), and a residual matrix (a matrix of residual values for each gene for each sample after subtracting the estimated cycle stage effect). We have also added an option to return the full set of residuals for every time point for all genes. This matrix of residual values can be used for additional ad-hoc analysis by the user.

We note that EndoTime (and the commercialised ERA test) does not provide a confidence score or interval. Rather, an asynchrony score is generated with the `endotime_train` function using the optimal time point for each gene and calculating the standard deviation between the times. This process can be replicated using the endest residual matrices output using all genes or, more preferably, a subset of informative genes, if the user chooses to do so. This would just involve taking the time point that generated the minimal residual for each gene of interest, then calculating a measure of variance with these time points. These residual values are subject to batch effects, so only intra-study comparisons should be performed when using the raw values. Asynchrony can also be checked by seeing if the MSE curve looks unusual with multiple local minima and no obvious single global minimum.

Even though statistical confidence estimates are not technically possible with our method, we reiterate an answer we submitted previously that demonstrates the robustness of our cycle stage estimates: *"How accurate these results are in terms of cycle stage is more difficult to ascertain. The comparison of NGS versus Illumina data from the same samples (Figure 4C) shows that at times of the cycle where gene expression is changing rapidly, such as menstruation and mid-secretory (see Figure 6a), there is very high concordance between the 2 different methods. This agreement suggests that results at these stages of the cycle are accurate to within less than 3-4% (approximately 1 day). At other times, such as during most of the proliferative phase, there is less agreement, suggesting an accuracy of around 7-11% (2-3 days). By contrast, pathology dating during the proliferative phase is limited to early-, mid- or late-proliferative."*

- "Conversely, if the 6 selected genes all have identical expression patterns, or do not change

expression much over the time of interest, then the predictive value of EndoGene will be reduced.”

Two very different things have been mixed together here:

1) genes that do not change much over time. Of course these will not carry much temporal information.

2) genes that do change readily but have the same temporal pattern. As each is measured independently these do provide independent information on timing, irrespective of similarity in their temporal pattern.

We apologise if these 2 points appeared ‘mixed’, they were not meant to be. What we said in our previous response to Reviewer #4 was:

“We agree with this comment. What we were trying to point out is that selecting the 6 most suitable genes to run EndoTime for different stages of the cycle is not straightforward. To get the best discrimination over time the ideal selection will include genes that are changing expression rapidly over the time of interest, with some increasing and some decreasing. Conversely, if the 6 selected genes all have identical expression patterns, or do not change expression much over the time of interest, then the predictive value of EndoTime will be reduced.”

To reiterate, the published version of EndoTime provides a time estimate in the luteal phase, using specified genes (6 genes in the Lipecki et al. paper), and needs LH data for each endometrial sample (information that is often not available in public datasets). The model is required to be trained on the user’s data.

Our method provides a complete solution for the whole cycle without further training and no additional information such as timing of the LH surge or other events relating to cycle stage required.

A COMPARISON BETWEEN ENDEST (OUR METHOD) AND ENDOTIME

We have undertaken a detailed in-house comparison of our model, Endest, with EndoTime. We came to the following conclusions about differences between the 2 packages:

- EndoTime estimates are anchored to a real measurement, LH+ values, whereas Endest uses a 0 to 100 scale built from assuming 236 samples were uniformly distributed in the cycle. We do have pathology staging, which we can estimate fuzzy boundaries of certain stages (e.g. secretory phase is approximately 58 to 100), but we would not be able to obtain a meaningful prediction interval (e.g. saying this sample falls between 44 and 51 in our model with 95% confidence) that could be tested empirically
- EndoTime requires batch information to be taken into account as it has a batch correction step whereas Endest doesn’t and is robust to batch effects (as we show in our reply above). This means Endest can accurately analyse a single sample from any laboratory while EndoTime would optimally require multiple samples across the LH+ range in order to estimate the batch effect for each gene. This can also cause issues if batch composition differs (e.g. if one batch contains samples that skew later in the LH+ range)
- EndoTime requires the model to be trained by the user whereas Endest comes pre-trained and includes coefficients in the package. The EndoTime package includes some training data (PCR data from the six genes in the paper) that could be used for training alongside the user’s data if they use the same genes. The EndoTime training process takes 10s of minutes to hours (depending on convergence time, number of genes, and other parameters), compared to the seconds to 10s of seconds that Endest takes to run.
- EndoTime uses a sliding window for their gene curves, calculating mean and variance in the window, whereas the Endest model was trained using penalised regression splines. Since EndoTime has variances, it can obtain sample likelihood estimates for each time point, whereas Endest minimises over a loss function (MSE), which is less statistically meaningful. EndoTime can calculate likelihood estimates because the training has been done by including

samples in the new data and this would correct for batches. Theoretically, EndoTime could generate confidence intervals, but this hasn't been done.

- We encourage reviewers to try using our package and try using EndoTime to really understand the difference in ease of use. We found EndoTime very difficult to use.
- In conclusion, EndoTime and Endest are different, and we believe that from an intellectual property perspective do not infringe on each other. There are advantages and disadvantages to each method.